# Assessing the impact of CO₂ equilibrated ocean alkalinity enhancement on microbial metabolic rates in an oligotrophic system

Laura Marín-Samper[1], Javier Arístegui[1], Nauzet Hernández-Hernández[1], Joaquín Ortiz[1], Stephen D. Archer[3], Andrea Ludwig[2], Ulf Riebesell[2]

[1] Instituto de Oceanografía y Cambio Global, Universidad de Las Palmas de Gran Canaria, 35017 Telde, Spain
[2] GEOMAR Helmholtz Centre for Ocean Research Kiel, 24148 Kiel, Germany
[3] Bigelow Laboratory for Ocean Sciences, 60 Bigelow Dr., PO Box 380, East Boothbay, Maine 04544, USA

*Correspondence to*: Laura Marín-Samper (laura.marin@ulpgc.es), Javier Arístegui (javier.aristegui@ulpgc.es)

**Abstract.** Ocean Alkalinity Enhancement (OAE) is a Negative Emissions Technology (NET) that shows significant potential for climate change mitigation. By increasing the bicarbonate ion concentration in ocean water, OAE could enhance long-term carbon storage and mitigate ocean acidification. However, the side effects and/or potential co-benefits of OAE on natural planktonic communities remain poorly understood. To address this knowledge gap, a mesocosm experiment was conducted in the oligotrophic waters of Gran Canaria. A CO₂-equilibrated Total Alkalinity (TA) gradient was employed in increments of 300 µmol·L⁻¹, ranging from ~2400 to ~4800 µmol·L⁻¹. This study represents the first attempt to evaluate the potential impacts of OAE on planktonic communities under natural conditions. The results show that Net Community Production (NCP), Gross Production (GP), Community Respiration (CR) rates, as well as the metabolic balance (GP:CR), did not exhibit a linear response to the whole alkalinity gradient. Instead, significant polynomial and linear regression models were observed for all rates up to ΔTA1800 µmol·L⁻¹, in relation to the Dissolved Inorganic Carbon (DIC) concentrations. Notably, the ΔTA1500 and 1800 µmol·L⁻¹ treatments showed peaks in NCP shifting from a heterotrophic to an autotrophic state, with NCP values of 4 and 8 µmol O₂ kg⁻¹ d⁻¹, respectively. These peaks and the optimum curve were also reflected in the nanoplankton abundance, size-fractionated chlorophyll *a* and ¹⁴C uptake data. Furthermore, abiotic precipitation occurred in the highest treatment after day 21 but no impact on the measured parameters was detected. Overall, a damaging effect of CO₂-equilibrated OAE in the range applied here, on phytoplankton primary production, community metabolism and composition could not be inferred. In fact, a potential co-benefit to OAE was observed in the form of the positive curvilinear response to the DIC gradient up to the ΔTA1800 treatment. Further experimental research at this scale is key to gain a better understanding of the short and long-term effects of OAE on planktonic communities.

**Keywords.** CDR, OAE, alkalinization, plankton, primary production, metabolic rates

## 1 Introduction

Limiting global warming to between 1.5°C and 2°C relative to preindustrial times, as stipulated in the 2015 Paris Agreement, will be necessary to avoid long-term, dangerous climatic consequences. Out of all the scenarios outlined in the fifth IPCC Assessment Report that meet this temperature target, 87% require extensive deployment of technologies to remove and sequester carbon dioxide ($CO_2$) from the atmosphere (Burns and Corbett, 2020; IPCC, 2018). Similarly, the Shared Socio-Economic Pathways that assume net-zero $CO_2$ emissions being reached by 2050, and negative emissions for the rest of this century, are the only ones in which the temperature increase is "*more likely than not*" bounded to below 2°C (IPCC, 2022). Besides, an estimated 26% of the anthropogenic $CO_2$ emitted between 1750 and 2020 has been taken up by the ocean through sea-gas exchange (Friedlingstein et al., 2022), subsequently altering its chemistry (Feely et al., 1985; Orr et al., 2005), a process that is commonly known as ocean acidification (OA). This phenomenon is notorious for being a threat to a wide range of marine taxa in terms of overall survival, calcification, growth, development and abundance (Kroeker et al., 2013, 2010; Wittmann and Pörtner, 2013; Hendriks and Duarte, 2010). The implementation of carbon dioxide removal (CDR) strategies will thus be crucial to timely offset the hard-to-abate emissions (Haszeldine et al., 2018; National Academies of Sciences, 2018; Renforth et al., 2013). Yet most approaches remain understudied, particularly those focused on ocean-based CDR (Gattuso et al., 2021, 2018; Rau et al., 2012).

Ocean alkalinity enhancement (OAE) is one of the ocean-based negative emissions technologies (NETs) that is presently being considered. It consists of atmospheric $CO_2$ removal by enhancing the ocean's carbon uptake capacity through mineral weathering (Kheshgi, 1995). It involves the dissolution of carbonate or silicate-based alkaline or alkali compounds/minerals in seawater, which consumes protons, altering the carbonate chemistry equilibrium by pushing it towards the carbonate and bicarbonate ion species. Thereby dissolved $CO_2$ concentration is reduced, counteracting OA while allowing for additional $CO_2$ uptake from the atmosphere. Model studies indicate that OAE could potentially remove between 3 and 10 Gt of atmospheric $CO_2$ per year (Feng et al., 2017; Harvey, 2008). Given the urgency to remove, capture, and store atmospheric $CO_2$ (Haszeldine et al., 2018; IPCC, 2018, 2022) and the ocean's potential to do so (Burns and Corbett, 2020), the evaluation of OAE applicability is of vital importance. Its implementation will depend on its scalability and on its environmental safety.

There are many proposed approaches for OAE deployment. For example, the supply of ground up minerals to coastal environments, the injection of alkaline solutions to, or dispersal of alkaline particles over, the surface ocean, and the electrochemical acid removal from seawater (Eisaman et al., 2023; Renforth and Henderson, 2017). In the present study we simulated a carbonate-based alkalinity addition to the open ocean surface, using a mesocosm approach. The waters off the coast of Gran Canaria were selected due to their oligotrophic nature (Supp. Fig. S1)

There are many model-based studies that focused on evaluating the feasibility, scalability and efficacy of this OAE approach (Butenschön et al., 2021; Caserini et al., 2021; González and Ilyina, 2016; Ilyina et al., 2013; Kheshgi, 1995; Lenton et al., 2018). But although conceptually it shows potential to mitigate OA and for CDR at global and regional scales, all the model simulations are based on a series of assumptions that remain poorly understood (Hartmann et al., 2023). This is due to the lack of focused experimental work under natural conditions.

Choosing a suitable approach to employ OAE is essential and complex. The maintenance of high alkalinity levels and thus the avoidance of alkalinity consumption through abiotic carbonate precipitation is key to ensure its CDR potential (Hartmann et al., 2023). Additionally, the type of source mineral (Bach et al., 2019), grinding fineness, whether it is added in its particulate form or in solution, if the latter is $CO_2$ equilibrated prior to addition or not, can all influence its potential environmental impacts. Also, precipitation occurrence may depend on the targeted TA level, especially if $CO_2$ sequestration is not undertaken prior to addition, but also on the presence of biogenic (Enmar et al., 2000; Nassif et al., 2005) or abiotic particles in seawater (Moras et al., 2022). Therefore, the latter may all impact its CDR efficiency. The simplest OAE deployment strategy is the direct dispersal of ground up minerals to the surface ocean (Harvey, 2008; Köhler et al., 2013). This method, however, may facilitate abiotic precipitation by supplying substrate for carbonate formation in an already supersaturated medium (Wurgaft et al., 2021). Additionally, if silicate-based (through the use of, for instance, dunite, an olivine rich mineral), it may cause the release of potentially harmful dissolution by-products such as trace metals (Bach et al., 2019; Ferderer et al., 2022; Montserrat et al., 2017; Meysman and Montserrat, 2017). Thus, despite being the simplest, it may not be the most suitable approach. The impacts on biota of different OAE strategies may also depend on the associated changes to the carbonate chemistry. This is especially true for non-$CO_2$ equilibrated OAE deployment scenarios where $p$$CO_2$ would be decreased, and thus pH more heavily altered than when employing an equilibrated approach (Bach et al., 2019; Paul & Bach, 2020; Chen et al., 1994; Giordano et al., 2005; Riebesell et al., 1993).

As a first attempt to evaluate OAE at a mesocosm scale, specifically to test the effect of the associated increment in total alkalinity (TA) and dissolved inorganic carbon (DIC), and to examine TA stability, we deployed an air equilibrated alkalinity gradient with carbonate-based solutions. Therefore, our TA manipulation did not contain any associated, and potentially harmful, dissolution by-products, nor the $p$CO2 was decreased. This way, simulating the alkalinity levels reached as one gradually moves away from a hypothetical OAE point source in oligotrophic conditions, under a best-case scenario. Changes in metabolic rates, primary production, chlorophyll *a* concentration and community composition, associated with the alkalinity gradient applied, were monitored. The goal was thus to detect possible environmental impacts and alkalinity thresholds. No major effects were expected since the carbonate chemistry parameters that are believed to drive phytoplankton growth, $CO_2$ and $H^+$ concentration (Paul & Bach, 2020), remained unaltered and moderately decreased respectively.

## 2 Materials and Methods

### 2.1 Experimental Design and Sampling

The experiment (KOSMOS Gran Canaria 2021) was set up at the Oceanic Platform of the Canary Islands' (PLOCAN) pier in the Taliarte harbour, Gran Canaria (Canary Islands, Spain), from September 14[th] to October 16[th], 2021. Nine mesocosms were deployed. They were supported by floating frames with joined flexible bags of 4m in length that were suspended and enclosed at the bottom with a conical sediment trap (Goldenberg et al., 2022; Supp. Fig. S2). Mesocosms were simultaneously filled up on September 10[th], 2021, with pre-filtered (3 mm) seawater pumped from nearby offshore waters (from the integrated water

column going from 2-12 m depth) with a peristaltic pump (14 m$^3$ h$^{-1}$, KUNZ SPF60, Flexodamp FD-50). Seawater was distributed equally across all mesocosms using a digital flow meter. The attained final volumes ranged between 8001 - 8051 L.

To examine the effects of an increment in TA, we applied a $CO_2$-equilibrated nine step alkalinity gradient in increments of 300 µmol · L$^{-1}$, with $Na_2CO_3$ and $NaHCO_3$ stock solutions. The latter were prepared by adding 22 kg of each salt separately to 22 kg of deionized water. The volume containing the difference in TA between the ambient and the target levels was added to each mesocosm. The applied gradient is displayed in Table 1. The averages of the measured (see section 2.2) TA and DIC are shown in italics, and were used to calculate the rest of the carbonate chemistry parameters using CO2SYS v2.1 software (Lewis and Wallace, 1998). Lueker et al.'s (2000) carbonate dissociation constants ($K_1$ and $K_2$), and the boron from Uppström (1974), were the constants employed for the mentioned calculation. The measured nutrient concentrations (Supp. Fig. S1) and *in situ* salinity were also used.

**Table 1. Averages ± the standard errors for the whole experiment after the Total Alkalinity (TA) manipulation (Day > 4), of the measured TA and Dissolved Inorganic Carbon (DIC), and of the theoretical values obtained through the CO2SYS v2.1 software for the rest of the carbonate system parameters, per mesocosm (MK), where "n" corresponds to the sample size. The TA, DIC, bicarbonate ($HCO_3^-$) and carbonate ($CO_3^{2-}$) concentrations are reported in µmol · L$^{-1}$, $p CO_2$ is in µatm, and pH is conveyed using the seawater scale.**

| MK | TA | DIC | pH$_{sw}$ | $p CO_2$ | $HCO_3^-$ | $CO_3^{2-}$ | $\Omega_{Ca}$ | $\Omega_{Ar}$ |
|---|---|---|---|---|---|---|---|---|
| **5** | *2427.7* | *2119.6* | 8.03 | 417.7 | 1889.6 | 219.4 | 5.2 | 3.4 |
| n = 15 | *± 4.31* | *± 2.92* | ± 0.003 | ± 2.87 | ± 2.59 | ± 1.48 | ± 0.03 | ± 0.02 |
| **1** | *2706.4* | *2348.5* | 8.06 | 435.6 | 2078.6 | 257.0 | 6.1 | 4.0 |
| n = 15 | *± 10.24* | *± 3.27* | ± 0.012 | ± 15.94 | ± 5.44 | ± 6.02 | ± 0.14 | ± 0.09 |
| **7** | *3003.7* | *2593.6* | 8.10 | 427.7 | 2271.0 | 310.0 | 7.3 | 4.8 |
| n = 15 | *± 7.59* | *± 4.73* | ± 0.006 | ± 6.34 | ± 5.60 | ± 3.94 | ± 0.09 | ± 0.06 |
| **4** | *3297.4* | *2829.8* | 8.14 | 429.8 | 2456.2 | 361.0 | 8.5 | 5.6 |
| n = 15 | *± 4.45* | *± 4.53* | ± 0.007 | ± 7.82 | ± 8.07 | ± 4.49 | ± 0.11 | ± 0.07 |
| **9** | *3603.9* | *3079.6* | 8.16 | 438.2 | 2654.1 | 412.6 | 9.8 | 6.4 |
| n = 15 | *± 7.27* | *± 4.95* | ± 0.005 | ± 5.40 | ± 6.18 | ± 4.03 | ± 0.09 | ± 0.06 |
| **3** | *3881.7* | *3295.7* | 8.19 | 435.2 | 2814.8 | 468.1 | 11.1 | 7.3 |
| n = 15 | *± 8.23* | *± 6.65* | ± 0.007 | ± 8.58 | ± 10.89 | ± 6.55 | ± 0.15 | ± 0.10 |
| **6** | *4165.4* | *3507.0* | 8.22 | 429.7 | 2969.3 | 528.5 | 12.5 | 8.2 |
| n = 14 | *± 7.77* | *± 6.43* | ± 0.007 | ± 8.13 | ± 11.40 | ± 6.92 | ± 0.16 | ± 0.10 |
| **2** | *4458.0* | *3752.3* | 8.23 | 443.7 | 3160.9 | 578.4 | 13.7 | 9.0 |
| n = 15 | *± 7.42* | *± 6.72* | ± 0.005 | ± 6.26 | ± 9.72 | ± 5.21 | ± 0.12 | ± 0.08 |
| **8** | *4655.8* | *3920.4* | 8.23 | 461.7 | 3299.0 | 607.8 | 14.4 | 9.4 |
| n = 15 | *± 22.09* | *± 13.53* | ± 0.007 | ± 8.53 | ± 9.99 | ± 9.15 | ± 0.22 | ± 0.15 |

Mesocosms were placed in order from 1 to 9 along the Taliarte pier, thus the actual TA treatments (from ΔTA 0 – 2400 µmol · L$^{-1}$, Table 1) were set out in random order. Custom-made samplers, constructed with 2.5 m long polypropylene tubing with a valve at each end, and a 5 L internal volume, were used to collect depth-integrated samples. These were collected every two days for a 33-day period. For further details on all activities conducted throughout the experiment including CTD and net tows, sediment trap pumping, mesocosm cleaning and overall maintenance, refer to Supp. Fig. S3.

## 2.2 TA and DIC measurements

TA and DIC samples were collected directly from the custom-made samplers into 250 mL glass flasks, allowing for substantial overflow and no headspace to avoid contamination. The samples were sterile filtered (0.2 μm, SARSTEDT, Nümbrecht, Germany) with a peristaltic pump. TA concentrations were determined by potentiometric titration using a Metrohm 862 Compact Titrosampler with HCl 0.05 M as the titrant, Aquatrode Plus (Pt1000), and 907 Titrando unit as in Chen et al., 2022. DIC concentrations were measured using an AIRICA system (Marianda, Kiel, Germany; see Gafar & Schulz, 2018, and Taucher et al., 2017) with a differential gas analyzer (LI-7000, LI-COR Biosciences GmbH, Bad Homburg, Germany) at room temperature and within 12 h.

## 2.3 Metabolic Rates through Oxygen Production and Consumption

Gross Production (GP), Net Community Production (NCP), and Community Respiration (CR) rates were determined by oxygen production and consumption in calibrated 125 mL nominal volume soda lime glass bottles following the Winkler method and the recommendations from Bryan et al. (1976), Carpenter (1966), and Grasshof et al. (1999). Polycarbonate bottles were filled with 4.5 L of seawater per mesocosm on each sampling day and brought to the lab. Out of these samples, twelve soda lime bottles per mesocosm were first rinsed with sample water and then randomly filled, allowing ample overflow, using a silicone tube with an attached 280 µm mesh on one end. The lids were then carefully placed, and each individual bottle was checked to be bubble free. Four subsamples per mesocosm were fixed at the moment of collection, "initials", through the addition of 1 mL of a manganese sulphate ($MnSO_4$) solution and 1 mL of a sodium iodide (NaI) based alkaline solution, in this order. They were later covered with a blackout piece of fabric and stored in a rack underwater. Another four bottles were incubated in the "dark", and the remaining four were incubated under "light" conditions. The "dark" ones were set inside light proof bags, which were then placed in a black opaque box. The "light" ones were randomly distributed inside clear methacrylate boxes, which were covered with a blue foil (172 Lagoon Blue foil, Lee filters, Burbank, USA) to better simulate the light spectrum of the water column. The boxes containing the light and dark bottles, and the rack with the initials, were placed in an outside pool found in the Parque Científico Tecnológico Marino of Taliarte, fed with a constant flow of seawater from the Taliarte pier. Data loggers (HOBO UA-002-64, Australia/New Zealand) were put inside the incubators to monitor the temperature (approximately 24.3 and 23.8 °C during the day and night, respectively) and light (ranging from 0.25 to approximately 2313.15 µmol photons $m^{-2}$ $s^{-1}$) conditions throughout the experiment. After an incubation period of 24 hours, all samples were fixed and left to sediment for at least 2 hours. Finally, samples were acidified with 1 mL of 5 M sulphuric acid ($H_2SO_4$) right before being analysed with an automated titration system, with colorimetric end-point detection (Dissolved Oxygen Analyzer, SIS Schwentinental, Germany), using a 0.25 M sodium thiosulphate solution ($Na_2S_2O_3 * 5H_2O$) as the titrant. The mean of each set of four replicates was used to calculate CR, NCP, and GP rates, using the following Eq. (1), Eq. (2) and Eq. (3) respectively:

$$CR\ [\mu mol\ L^{-1}h^{-1}] = \frac{Conc_I - Conc_D}{h_D} \tag{1}$$

$$\text{NCP } [\mu\text{mol L}^{-1}\text{h}^{-1}] = \frac{\text{Conc}_L - \text{Conc}_I}{h_L} \tag{2}$$

$$\text{GP } [\mu\text{mol L}^{-1}\text{h}^{-1}] = \text{CR} + \text{NCP} \tag{3}$$

where $\text{Conc}_I$, $\text{Conc}_D$ and $\text{Conc}_L$ correspond to the mean oxygen concentration of the initial, dark, and light samples, respectively. $h_L$ and $h_D$ stand for incubation time in hours under light and dark conditions, respectively. The metabolic balance was later calculated by dividing the obtained GP by CR.

## 2.4 Size-fractionated Primary Production through $^{14}$C uptake

Samples from each mesocosm were taken into 10 L High Density Polyethylene (HDPE) canisters and transported to the GOB laboratories every two sampling days. Primary production (PP) in pico (0.2-2 µm), nano (2-20 µm), and micro (20-280 µm) size fractions were measured following a modified version of the approach by Cermeño et al. (2012). Four culture flasks (Sarstedt TC Flask d15, Nümbrecht, Germany) per mesocosm were filled up to the bottle neck (70 mL) and spiked with 80 µL (0.296 MBq) of a $^{14}$C-labeled

sodium bicarbonate (NaH$^{14}$CO$_3$, Perkin Elmer, Waltham, USA) stock solution (3.7 MBq mL$^{-1}$). Prior to $^{14}$C inoculation, samples were prefiltered through a 280 µm mesh to exclude most of the zooplankton fraction. Each flask was then closed and gently homogenized. All culture flasks were incubated for 24 h in an environmental chamber (Aralab FitoClima 600 Bio Chamber, Lisbon, Portugal) at *in situ* light (12 h light-dark cycle with a mean daily PAR intensity of ~500 µmol photons m$^{-2}$ s$^{-1}$) and temperature (21 - 24 °C

depending on the temperature in the mesocosms on each sampling day). One out of the four culture flasks per mesocosm was incubated inside a light-proof bag to prevent photosynthesis.

After incubation, all samples were sequentially filtered on a circular filtration manifold (Oceomic, Fuerteventura, Spain) under low vacuum pressure (<200 mbar) through polycarbonate membrane filters with pore sizes of 20 µm (top), 2 µm (middle) and 0.2 µm (bottom) (DHI GVS 20 µm, Hørsholm, Denmark,

Whatman Nuclepore 2 µm & 0.2 µm, Maidstone, UK). The manifold allowed to collect the filtrate in 120 mL HDPE bottles. The filters were placed in 5 mL scintillation vials (Sarstedt HDPE Mini-vial, Nümbrecht, Germany), while 5 mL of the filtrates were transferred to 20 mL scintillation vials (Sarstedt HDPE Scintillation vial, Nümbrecht, Germany) for dissolved organic carbon production (PP$_{DOC}$) determination. To remove the remaining inorganic $^{14}$C, all samples were acidified. To do so, the 5 mL vials with the filters

were placed inside a desiccator and exposed to fuming hydrochloric acid (HCl 37 %) for 24 h. Whilst 100 µL of hydrochloric acid (HCl 17.5 %) were added to the filtrate subsamples and placed on an orbital oscillator at 60 rpm, also for 24 h.

After acidifying, filters were pushed into the vials and 3.5 and 10 mL of scintillation cocktail (Ultima Gold XR, Perkin Elmer, Waltham, USA) were added to the filters and the liquid samples, respectively. All vials

were vigorously shaken and left for an additional 24 h in the dark before being measured with a scintillation counter (Beckman LS-6500, Brea, USA). The counted disintegrations per minute (DPM) were used to calculate primary production rates [µg C L$^{-1}$ h$^{-1}$] using the following Eq. (4):

$$\text{PP} = \frac{V_S}{V_F} \cdot \frac{\text{DIC} \cdot (\text{DPM}_S - \text{DPM}_D)}{\text{DPM}_A \cdot t_i} \tag{4}$$

where $V_S$ = sample volume (L), $V_F$ = filtered volume (L), $DPM_S$ = sample disintegrations per minute, $DPM_D$ = dark-incubated sample disintegrations per minute, DIC = dissolved inorganic carbon ($\mu$mol C L$^{-1}$), $DPM_A$ = initially added $^{14}$C in disintegrations per minute, and $t_i$= time of incubation (h).

The average of the triplicates was used to calculate the final PP rates. The three size fractions were summed up to calculate the particulate organic carbon production ($PP_{POC}$). Moreover, the Eq. (5) below was utilized to calculate the Percentage of Extracellular organic carbon Release (PER):

$$PER\ (\%) = \frac{PP_{DOC}}{PP_{POC} + PP_{DOC}} \cdot 100 \tag{5}$$

## 2.5 Size-fractionated Chlorophyll *a*

Chlorophyll *a* (Chl*a*) samples for each mesocosm were collected into 500 mL dark bottles from the same 10 L canisters as PP. Samples were sequentially filtered through superimposed Polycarbonate filters of 20 $\mu$m, 2 $\mu$m and 0.2 $\mu$m pore size (DHI GVS 20 $\mu$m, Hørsholm, Denmark, Whatman Nuclepore 2 $\mu$m and 0.2 $\mu$m, Maidstone, UK). The filters were stored at -20 °C while pending analysis. The pigment was extracted by submerging the filters in 10 mL of acetone (90%) at -20 °C for 24 h. The extracts were analyzed using a benchtop fluorometer Turner Design AU-10 (San Jose, USA) as in Welschmeyer (1994). Total Chl*a* concentration was determined by adding up the three size fractions.

## 2.6 Prokaryotic and Eukaryotic Abundances

Duplicate flow cytometry samples were collected every two days and ran in vivo. A CytoSense (Cytobuoy, Woerden, Netherlands) flow cytometer was used, and the default software (Cytoclust) was employed to differentiate the phytoplankton population clusters based on red, yellow, and green fluorescence as well as forward and side scatter, which are indicators of size and cell complexity (Dubelaar and Gerritzen, 2000). *Synechococcus* and picoeukaryotes fall within the same forward and side scatter range, but *Synechococcus* are distinguished due to their yellow fluorescence content. Picoeukaryotes and nanoeukaryotes both contain red fluorescence, but the latter group are larger in size and complexity. Thus, falling within distinct forward and side scatter ranges.

## 2.7 Data Analysis

The experiment was divided into two phases (I: days 5-19, II: days 21-33; the reasons for this division are explained in the results section). All parameters were analyzed in relation to the alkalinity gradient deployed using simple linear regressions. Additionally, in the parameters that showed a potential curvilinear trend in relation to the TA/DIC gradient, linear and polynomial regression models were also fitted excluding the two highest treatments. For these parameters, in order to avoid over-fitting, cross validation was used to assess the polynomial model's performance to pick the best-fitting model order. DIC was chosen as the predictor variable for the latter. Averages of the response variables for each phase and for the entirety of the experiment, in both cases excluding the days prior to the TA addition, were used. Assumptions of normality were tested using q-q and Shapiro-Wilk tests on test residuals. Data analyses were performed using RStudio (2022.02.3 Build 492; packages stats, ggplot2 v3.3.5;Wickham et al., 2016).

## 3 Results

### 3.1 Carbonate chemistry temporal development and phase determination

The TA gradient in increments of 300 µmol · L$^{-1}$ was attained, and DIC and TA were stable up to day 21 (Figure 1.A). The experimental period up to that day, 5 – 19, was differentiated and designated as phase I (< 14 days after the TA addition), with phase II defined as the period starting on day 20 that coincided with an abrupt change in the biological response among the mesocosms. Additionally, in this second phase (>14 days after the TA addition), indirect abiotic precipitation occurred in the highest treatment, Δ2400 µmol · L$^{-1}$. Precipitates were visibly forming on the mesocosm walls by day 28, a process that advanced quite rapidly during the 6 days after cleaning. The precipitation process lasted until the end of the experiment and led to a TA and DIC loss of ~ 293.7 µmol · L$^{-1}$ and 175.3 µmol· L$^{-1}$, respectively (Figure 1.A and B).

After the alkalinity addition on day 4, the pH varied slightly according to the gradient applied ranging from 8.03 in the control to almost 8.3 in the highest treatment (Figure 1.C). CO$_2$ partial pressure did not vary alongside the TA gradient due to the equilibrated nature of the alkalinity manipulation (Figure 1.D). However, the estimated $p$CO$_2$ in the highest treatment in phase II increased from ~ 450 in phase I, to a maximum of ~550 µatm by day 33 due to the triggered calcification process (ΔTA 2400 µmol · L$^{-1}$). It was ~50 µatm higher than the rest of the treatments starting on day 27, increasing towards the end of the experiment, when it was ~100 µatm greater than ambient levels (Figure 1.D). Because of the increase in $p$CO$_2$ in this treatment in phase II, pH dropped from 8.24 on day 18, down to a minimum of 8.16 at the end of the experiment (Figure 1. C and D).

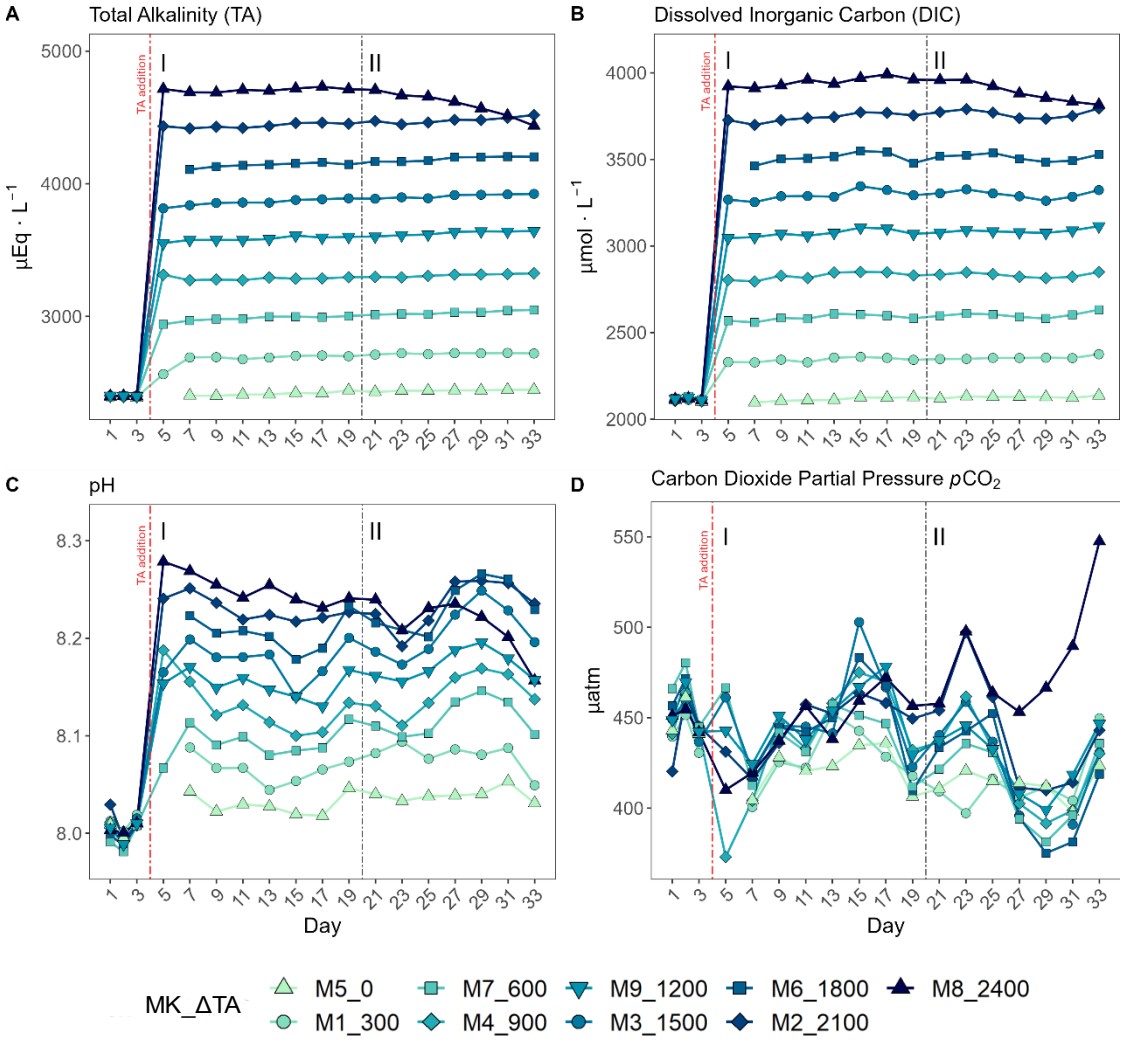

**Figure 1. Temporal development of A) Total Alkalinity (TA), B) Dissolved Inorganic Carbon (DIC), C) pH (seawater scale) and D) $p$CO$_2$ throughout the entire experiment for each mesocosm (MK) and treatment ($\Delta$TA). The x axis represents the number of days elapsed since the beginning of the experiment.**

### 3.2 Primary production and metabolic balance

As previously mentioned, another reason for the delineation of the mentioned phases (I: days 5-19, II: days 21-33) is the observed increase in production and chlorophyll-a concentration in specific intermediate treatments after day 20 when compared to phase I (Figure 2). This division of the experimental period was chosen to facilitate the system's response interpretation. Overall, NCP, GP, and the metabolic balance (GP:CR) show similar developments. All metabolic rates behaved differently in the two phases (Figure 2.A, B and C). In the first phase, CR accounted for most of the GP, while NCP was for the most part negative (more respiration than oxygen production; Figure 2.B and D). In contrast, a peak in GP and NCP rates occurred at $\Delta$1500 and $\Delta$1800 during the second phase, showing 2 and 3-fold increased GP, respectively. Autotrophy was also observed in the $\Delta$600 and $\Delta$900 treatments during this phase although only for two days (Figure 2.B and D).

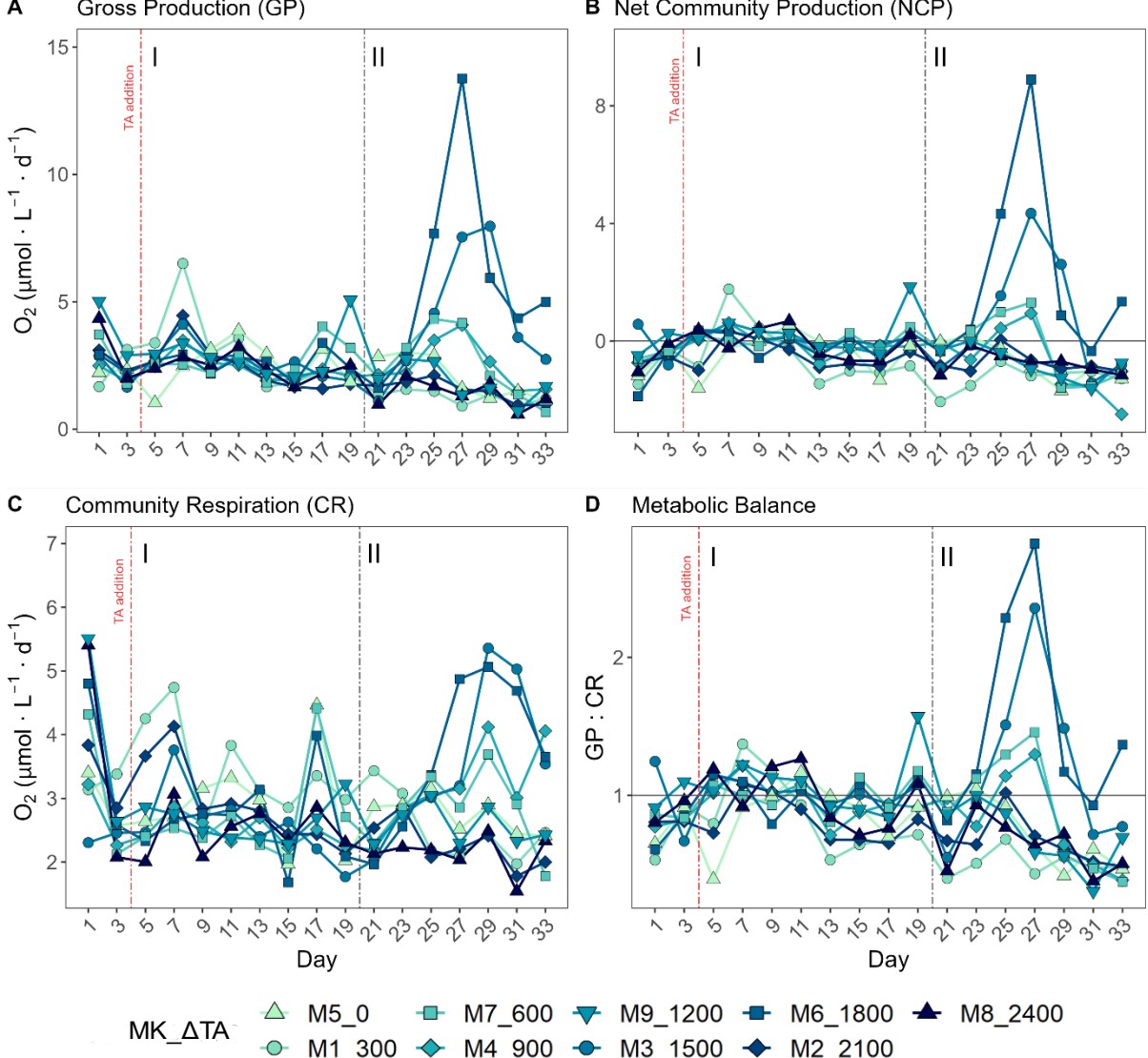

**Figure 2. Results for metabolic rates measured through oxygen production and consumption showing the temporal development of A) the gross production (GP), B) the net community production (NCP), C) the community respiration (CR), and D) of the metabolic balance (GCP over CR). In the legend, MK corresponds to Mesocosm and ΔTA to ΔTotal Alkalinity. The x axis represents the number of days elapsed since the beginning of the experiment.**

Phase-averaged linear regressions with the whole TA gradient revealed no significant treatment effect
(alpha < 0.05) on NCP, GP, and CR rates as well as metabolic balance (GP:CR) (Supp. Fig. S4).
Additionally, no impact of the abiotic precipitation in the highest treatment was observed regarding GP,
NCP, CR, GP:CR (Figure 2), $^{14}$C primary production, and Chl*a* concentration (Figure 3 and 4).

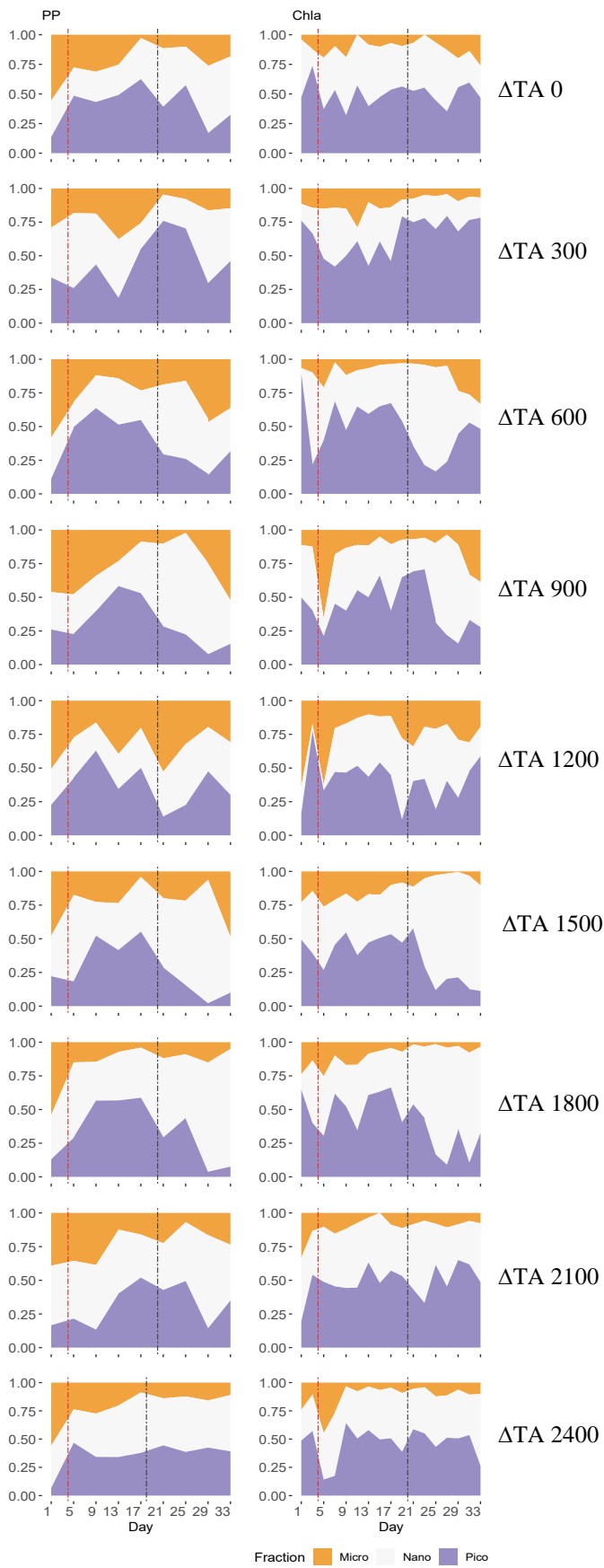

**Figure 3. Temporal development of the three phytoplankton size fractions' (pico 0.2-2 µm; nano 2-20 µm; micro >20 µm) relative contributions to primary production (PP) through $^{14}C$ uptake (left column) and chlorophyll a (Chl*a*) concentration (right column), for each treatment. The x axis represents the number of days elapsed since the beginning of the experiment.**

In terms of the relative contributions of pico, nano and micro to total PP and Chl*a*, differences between phases, although they are not statistically significant, they are only visually clear in the Δ600, Δ900, Δ1500 and Δ1800 treatments, those that showed autotrophy (Figure 2D) in the second phase (Figure 3). In these mesocosms, pico, and in Δ900 micro also in terms of PP (Figure 3, left), contributed the most in the first phase. However, in the second phase, nano became more dominant in these intermediate treatments, especially in Δ1500 and Δ1800. In Δ1200 the latter pattern is not clear. Instead, the micro fraction contributed more throughout the whole experiment when compared to all other treatments. Total PP and Chl*a* concentration data matched the spike in oxygen production observed in treatments Δ1500 and Δ1800, and also the slight increases found in treatments Δ600 and Δ900 (Figure 4). Data for PP on day 27, when oxygen production and Chl*a* concentration in Δ1800 was the highest of all values recorded throughout the entire experiment and for all mesocosms, were not collected. Meaning thus, the peak in Δ1800 reflected by the Chl*a* (Figure 4A and B), and the GCP and NCP rates (Figure 2), was excluded (Figure 4C and D). This explains why the peak in production is lower in the Δ1800 treatment than in the Δ1500, particularly in Figure 4D.

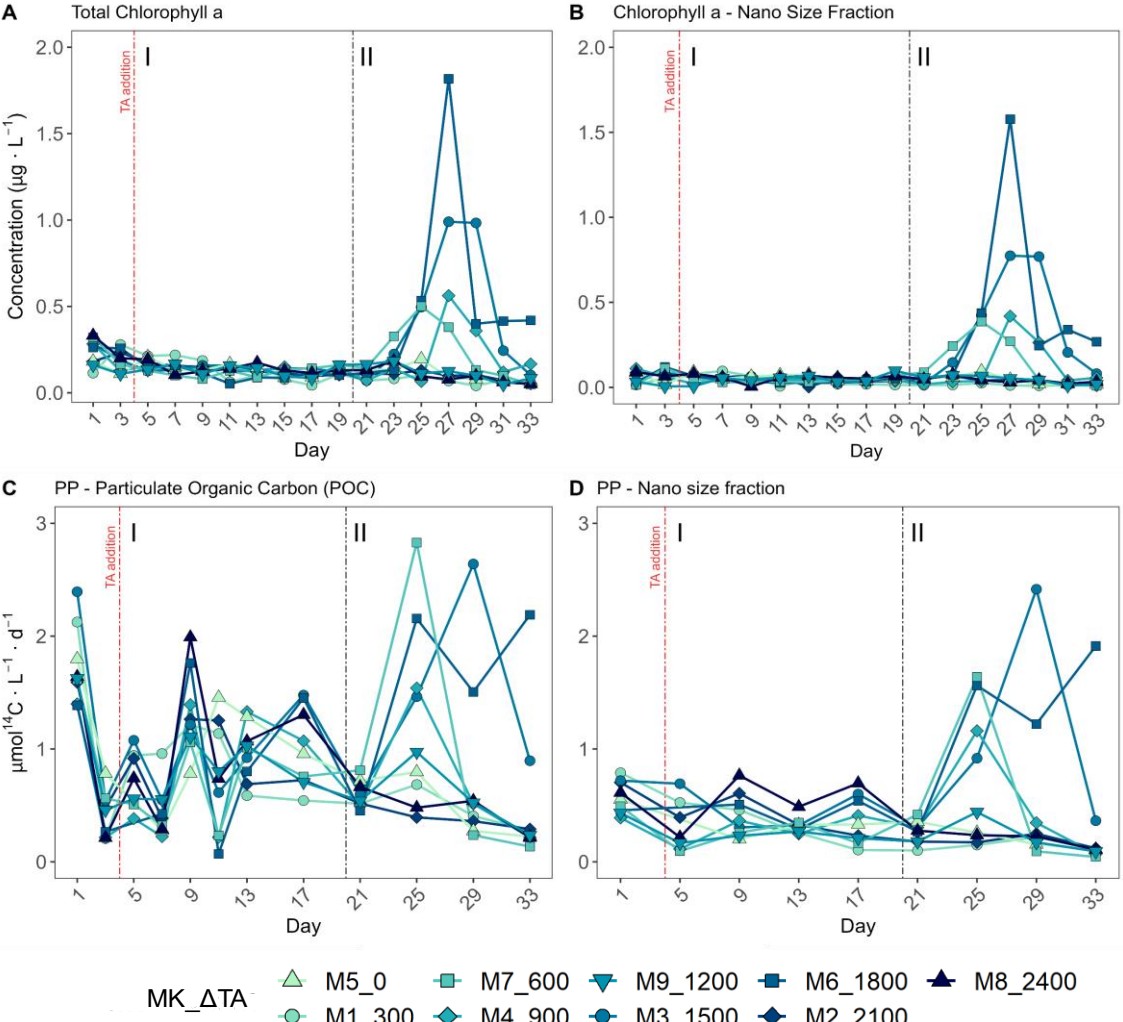

**Figure 4. Temporal development of A) total chlorophyll *a* (Chl*a*) concentration, B) the nano size fraction's (2 – 20 μm) contribution to the total Chl*a* concentration, C) total particulate organic carbon production (where PP is Primary Production), and D) of the nano size fraction's (same size range as for the Chl*a*) contribution to total PP. MK corresponds to Mesocosm and ΔTA to ΔTotal Alkalinity. The x axis represents the number of days elapsed since the beginning of the experiment.**

The increases in production observed in phase II were driven by nanoplankton growth (Figure 3, 4.B & D).
When considering all treatments, in phase I this size fraction showed a positive linear trend in relation to the alkalinity ($R^2 = 0.51$; $p = 0.031$) and the DIC ($R^2 = 0.50$; $p = 0.031$) gradients in terms of $^{14}C$ uptake. However, this significant relationship vanished by phase II.

Regarding Percent of Extracellular organic carbon Release (PER), no statistically significant linear relationship with the whole DIC gradient, chosen since it likely was the driver behind a potential response in PP rather than TA, was found (Figure 5). Moreover, and as is true for all other parameters presented in this study, PER behaved disparately during the two phases. For the intermediate treatments, where there was autotrophy in the second phase, PER values dropped in comparison to the two highest and two lowest treatments, while Δ1200 stayed the same. Additionally, if the two highest treatments are excluded from the analysis, a significant negative relationship between the PER and the DIC gradient can be observed (Figure 5).

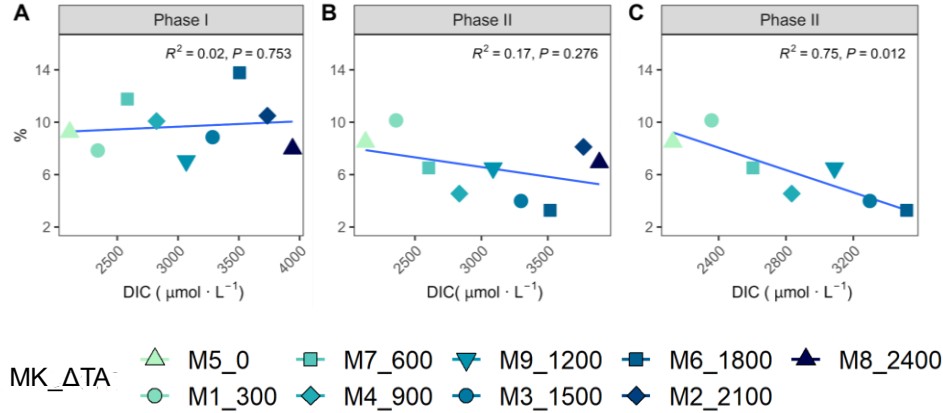

**Figure 5. Linear regressions between ΔTA treatment and PER per phase (A. phase I and B. phase II), and (C) removing the two highest treatments in phase II. MK corresponds to Mesocosm and ΔTA to ΔTotal Alkalinity.**

### 3.3 Pico- and Nano- eukaryote abundances

The second phase of the experiment was characterized by an increase in production and Chl*a* concentrations in all intermediate treatments below the two hishest and above the two lowest treatments, except Δ1200. While phase I was distinguished by extremely low GP, NCP, PP rates and Chl*a* throughout and across all mesocosms. Picoeukaryote abundance decreased during the first phase and picked back up 3-fold in the intermediate treatments going from Δ600 to Δ1800 (Figure 6A). *Synechoccocus* proliferated in phase II in the lower intermediate treatments (treatments Δ600 and Δ900) as seen in Figure 6B. Two nanoeukaryote groups could be distinguished based on complexity and red fluorescence content. Nanoeykaryotes(2) were larger, and contained more red fluorescence than the nanoeukarytes(1), and they also held some yellow fluorescence. Nanoeukaryote (1) abundance, despite gradually dropping throughout the experiment (Figure 6C), showed a positive linear relationship ($R^2 = 0.634$, p= 0.01) with TA across both phases. Nanoeukaryote (2) abundance drove GP, NCP, and PP rates, and contributed the most to Chl*a* in the intermediate treatments, except Δ1200, during phase II (Figure 4 and 6D). In addition, no impact of the indirect abiotic precipitation that occurred in the highest treatment during phase II was observed on any of the population abundances monitored (Figure 6). In fact, abundances of all groups in the latter treatment are comparable to those observed in the control.

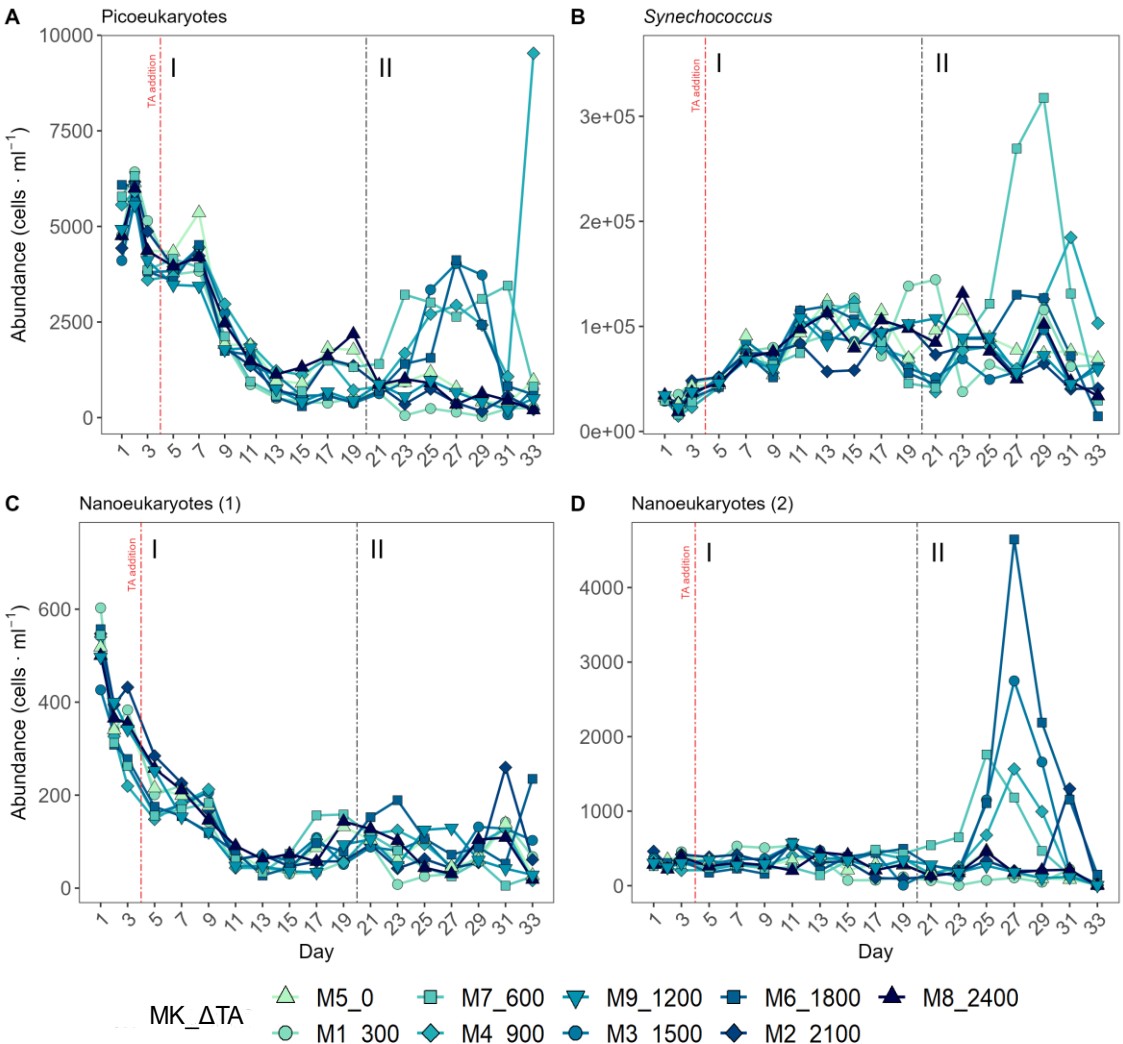

**Figure 6. Abundance in cells m⁻¹ obtained through flow cytometry of A) picoeukaryotes, B) *Synechococcus*, C) nanoeukaryote (1) and D) nanoeukaryote (2). The two latter correspond to two different nanoeukaryote populations by both complexity/size and red fluorescence content. MK corresponds to Mesocosm and ΔTA to ΔTotal Alkalinity. The x axis represents the number of days elapsed since the beginning of the experiment.**

### 3.4 Non-linear response vs no response

TA and DIC, in an equilibrated OAE approach, vary together (as TA increases, so does DIC; Supp. Fig. S5) and, if a potential non-linear response between the metabolic parameters listed in Table 2 were to be considered, the driver behind these relationships would most likely be DIC (key substrate for carbon fixation; Badger et al., 1998), not TA. The non-linear response was detected for the longer-term phase, meaning the averaged-out values of phase II, but also for the entire duration of the experiment (Figure 7). Average GP and NCP rates, GP:CR, total PP, Particulate and Dissolved Organic carbon (POC and DOC) production, and Chl*a*, the nanoplankton contribution to the latter two, and the nanoeukaryote abundances, all exhibited a gradual increase in the intermediate treatments, and a decline beyond ΔTA 1800 µmol · L⁻¹, during the mentioned time periods. Indeed, if the two highest treatments are excluded from the model, significant linear relationships emerge between DIC, and all the parameters listed above for both phase II and the entire experiment (Table 2). However, it is worth noting that these relationships yield stronger regression coefficients when second-order polynomial regression models are employed instead (Table 2).

Furthermore, a significant relationship is observed in phase I between NCP, metabolic balance and nanoeukaryote (2) abundance if the polynomial model is fitted (Table 2, left). Linear relationships also 360 become evident for the latter parameter when analyzed independently (Table 2, middle).

However, when linear regressions are employed and the two treatments with the highest responses ($\Delta$1500 and $\Delta$1800 µmol · L$^{-1}$) are excluded instead, the significance of all the previously described relationships is no longer observed (Table 2). Although, even when these two intermediate treatments are excluded, the nanoplankton contribution to PP in phase I, and the nanoeukayote (1) abundance throughout the experiment, 365 continue to exhibit a significant linear trend. This suggests that these specific relationships remain robust and significant, regardless of the exclusion of the highest-response treatments.

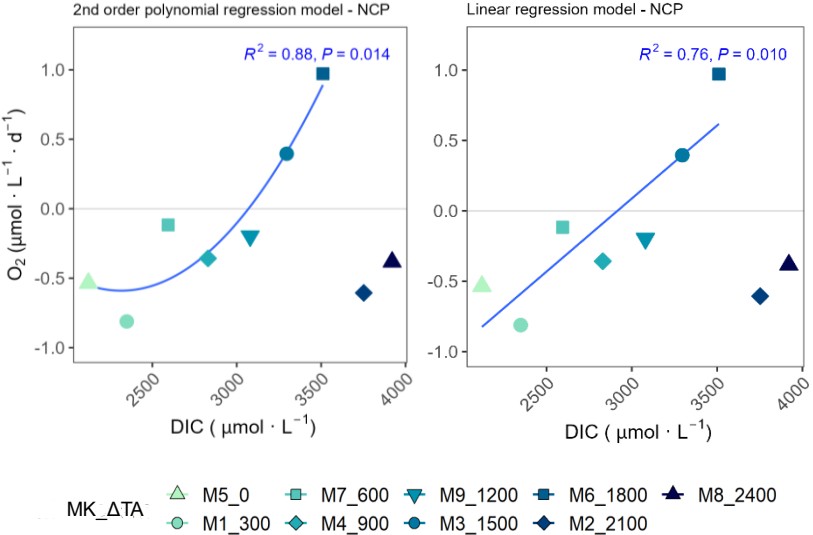

**Figure 7. Second order polynomial (left) and linear (right) regression models fitted to the treatments of up to $\Delta$TA1800 µmol · L$^{-1}$ relating Net Community Production (NCP) Rates and the associated Dissolved Inorganic Carbon (DIC) levels averaged out for the whole experiment. In the legend, MK corresponds to Mesocosm and TA to Total Alkalinity.**

**Table 2.** Summary tables showing the regression coefficient values of (left) second order polynomial regression models, (middle) linear regression models, in both cases excluding the two highest treatments (Δ2100 and Δ2400 μmol · L⁻¹), and (right) linear models excluding the two treatments that show the highest response (Δ1500 and Δ1800 μmol · L⁻¹), fitted for gross and net community production (GP and NCP), metabolic balance (GP:CR), $^{14}$C Primary Production (PP-total, POC and DOC), the nanoplankton fraction contribution to PP_Total (PP_Nano), total Chl*a* concentration (Chl*a*), the nanoplankton fraction contribution to total Chl*a* concentration (Chl*a*_Nano), and the abundances of nanoeukaryotes (1) and (2) counted through flow cytometry, in relation to DIC. The p-values are indicated by the symbol below each regression coefficient (see legend). All significant regressions are marked in bold letters.

| y | Polynomial without 2 highest treatments [y ~ DIC + I (DIC^2)] | | | Linear without 2 highest treatments [y ~ DIC] | | | Linear without Δ1500 and Δ1800 treatments [y ~ DIC] | | |
|---|---|---|---|---|---|---|---|---|---|
| | **Phase I** | **Phase II** | **Throughout** | **Phase I** | **Phase II** | **Throughout** | **Phase I** | **Phase II** | **Throughout** |
| GP | 0.034 | **0.834** * | **0.880** * | 0.007 | **0.630** * | **0.677** * | 0.192 | 0.158 | 0.405 |
| NCP | **0.729** • | **0.857** * | **0.881** * | 0.487 | **0.614** * | **0.763** * | 0.025 | 0.004 | 0.018 |
| GP:CR | 0.**758** • | **0.812** * | **0.878** * | 0.463 | **0.616** * | **0.767** ** | 0.016 | 0.022 | 0.0005 |
| PP_Total | 0.527 | **0.705** • | **0.791** * | 0.002 | **0.618** * | **0.532** • | 0.091 | 0.130 | 0.001 |
| PP_POC | 0.538 | **0.703** • | **0.806** * | 0.008 | **0.620** * | **0.561** • | 0.085 | 0.118 | 0.005 |
| PP_DOC | 0.250 | **0.792** * | **0.749** • | 0.160 | **0.656** * | **0.528** • | 0.057 | 0.104 | 0.002 |
| PP_Nano | 0.588 | **0.752** • | **0.788** * | 0.369 | **0.644** * | **0.371** * | **0.474** • | 0.048 | 0.096 |
| Chl*a* | 0.176 | **0.783** * | **0.782** * | 0.128 | **0.667** * | **0.668** * | 0.110 | 0.039 | 0.029 |
| Chl*a*_Nano | 0.243 | **0.779** * | **0.785** * | 0.148 | **0.653** * | **0.643** * | 0.039 | 0.028 | 0.023 |
| Nano (1) | 0.132 | **0.872** * | **0.687** • | 0.003 | **0.598** * | **0.577** * | 0.407 | 0.194 | **0.623** * |
| Nano (2) | **0.723** • | **0.658** • | **0.696** • | **0.683** * | **0.576** * | **0.623** * | 0.015 | 0.0001 | 9.86E-06 |

| **p-value** | 0 - 0.001 *** | 0.001 - 0.01 ** | 0.01 - 0.05 * | 0.05 - 0.1 • | 0.1 - 1 |
|---|---|---|---|---|---|

The cross-validation test results indicate that the second polynomial term of DIC is marginally significant (p-value between 0.05 and 0.1) while the first polynomial term (the linear model) is statistically significant (p-values < 0.05). Although notably, both of these terms have a positive coefficient. Thus, suggesting that, even though the second polynomial term leads to higher regression coefficients, it may have a weaker although albeit potentially relevant effect (still >90 % confidence level) on the response variable when compared to the linear model.

## 4. Discussion

The main goal of this study was to simulate an carbonate-based OAE scenario. As a first step, carbonate-based, $CO_2$ equilibrated solutions were used in order to simulate a best-case scenario. $CO_2$ equilibration, i.e., keeping $p$$CO_2$ levels constant, allows for greater alkalinity additions before the $CaCO_3$ saturation threshold is reached. The levels of exposure experienced by the microbial community (< 280 µm in the current study) at an alkalinity dispersal plume, were simulated through the ΔTA gradient. The oligotrophic waters surrounding the Canary Islands were chosen as an open ocean oligotrophic system analog in terms of nutrient availability and community composition.

A neutral response of the measured metabolic rates, PP, Chl$a$, and community composition, when taking the entire alkalinity range (from ambient to ~ 4600 µmol kg$^{-1}$) applied here into account, was observed. These results are consistent with four-day microcosm experiments carried out at sea with two natural microbial communities of the North Atlantic subtropical gyre (Subhas et al., 2022). In this case, only 3 alkalinity treatments were deployed, with the highest being ~ 4500 µmol kg$^{-1}$, and also using $NaHCO_3$ and $Na_2CO_3$ stock solutions. No major effect on the estimated net primary production, minor effects on community composition, and no influence on net calcification rates, were observed after 4 days. Results that, followed by those obtained from the current longer-term study, suggest this OAE approach may not entail significant alterations to microbial communities in oligotrophic pelagic systems.

However, nutrient limitation (Supp. Fig. S1) may have concealed more apparent responses to the TA and DIC gradients. Research on OAE's potential impacts in oligotrophic systems at a comparable scale is non-existent. In eutrophic environments, a transient positive impact on calcifiers, if present at the time of deployment, has been hypothesized due to the provision of additional substrate for calcification in the form of carbonate ions (Bach et al., 2019). Notably at and surrounding the alkalinity addition hotspot where the carbonate system is altered the most. Nevertheless, a recently published study showed no response of *Emiliania huxleyi* to a limestone inspired alkalinity addition in a laboratory setting, with high nutrient

availability, in terms of growth rates and elemental ratios after 6 days (Gately et al., 2023). Whether this is the case in a natural environment, and for longer term exposure to such conditions, is unknown. These results suggest that the effects of OAE on community structure and composition may be more complex than anticipated with the "green vs white" ocean hypothesis (Bach et al., 2019). However, further experimental research is necessary to evaluate the consequences of, for instance, a silicate- versus a carbonate-based OAE deployment but also of OAE in more eutrophic environments. More specifically regarding community structure, calcification and silicification, but also primary production and metabolic balance, to address key knowledge gaps.

**4.1 Potential for non-linear effects of OAE on metabolic rates**

In the current study, a linear short-term response was observed for the nano fraction's contribution to PP, and a positive relationship between nanoeukaryote (1) abundance and TA was detected when considering the averages for the whole experiment. These results are not entirely supported by those obtained by Ferderer et al. (2022). In their study, the water enclosed in ~55 L microcosms for 25 days was rich in inorganic nutrients ($PO_4^{3-}$ 0.79 ± 0.01 µM; $NO_x$ 6.38 ± 0.19 µM; 9.65 ± 0.39 µM $Si(OH)_4$). The alkalinity addition in their equilibrated treatment was of roughly 500 µmol · L$^{-1}$, which is comparable to the Δ600 µmol · L$^{-1}$ treatment in the present study which mildly responds in the second phase. They observed a significant difference in the Chl*a* concentrations, also driven by nanoplankton growth, between the control and their equilibrated alkalinity treatment in which the latter was lower. They report on a short-term response to the initial nutrient concentrations after closure. In the current study however, relationships with a gradient rather than differences between one treatment and another are reported on instead. Additionally, production and Chl*a* responses here occur in the long term, past day 27[th], and the significant linear trends stated at the beginning of this section are not with regard to Chl*a*. Furthermore, in the present study, no significant linear relationships with the whole TA gradient were found in any other parameter after a month-long exposure to such conditions.

Albeit at constant $p$CO$_2$, prolonged exposure to higher calcite/aragonite saturation states and moderate pH increases in an OAE dispersal plume has been hypothesized to lead to nonlinear and/or threshold-like responses in the long-term (Subhas et al., 2022). This pattern was noticed for the parameters listed in Table 2 in relation to DIC, suggesting there may in fact be an optimum curve-like response, and a threshold between Δ1800 and Δ2100 µmol L$^{-1}$ treatments (Figure 6). Indeed, if only the treatments below Δ2100 µmol L$^{-1}$ are considered, positive significant relationships, described by polynomial (higher regression

coefficients) and linear regression models (Table 2 and Figure 7), arise between DIC and NCP, GP, GP:CR,
PP-total, POC and DOC production, PP-Nano, total Chl*a,* Chl*a*-Nano, and both nanoeukaryote clusters'
abundances, in addition to the opposite pattern being reflected by the PER (Figure 7). PER is known to be
higher in oligotrophic than in eutrophic waters (Chróst, 1983; Teira et al., 2001). The observed significant
decrease in the PER associated to the OAE treatment up to $\Delta$1800 µmol L$^{-1}$ in the second phase suggests
there may have been a slight increase of inorganic nutrients in relation to the TA manipulation, potentially
caused by enhanced nitrogen cycling. In fact, the latter process is known to be pH dependent (Beman et al.,
2011; Fumasoli et al., 2017; Pommerening-Röser and Koops, 2005). The previous explanation may be
further supported by the increase in $NO_x$ ($NO_3 + NO_2$; in particular of nitrite, $NO_2^-$) concentrations (Sup.
Fig. S1) observed in the second phase. Additionally, Paul et al., (2024) observed a positive relationship of
particulate organic carbon to particulate organic nitrogen (POC: PON) ratios in the second phase, and a
negative relation of Particulate Organic Nitrogen (PON) concentrations in the first phase, both with the
OAE treatment during the same mesocosm study. Thus, considering that all the earlier described responses
occurred in the second phase (designated as long term), the peak in production may have been possible due
to a slight increase in heterotrophic turnover of organic nitrogen associated with the carbonate chemistry
manipulation, which would explain the lag in the observed responses.

Actually, considering the nutrient-depleted nature of the system in all mesocosms, the occurrence of the
peaks in production that drive the optimum curve-like relationship was unexpected. All that is currently
known about the species responsible for the increase in productivity observed in the four intermediate
treatments (Figure 6) is that it was a *Chrysochromulina* spp after exemplary samples having been analyzed
via microscopy. In a study carried out from May to June 1988 in the Kattegat, *C. polylepis* was monitored
prior to the decline of a bloom (Kaas et al., 1991). The authors measured its distribution, primary
production, and nitrogen dynamics and found that *C. polylepis* showed high affinity for ammonia. It was
its main nitrogen source and, as previously stated, the only nutrient that was not measured in the current
study.

An alternative explanation could be that the protagonist in the intermediate treatments during the second
phase of the experiment was *C. parkeae,* which is a life cycle stage of *Braarudosphaera bigelowii* (Suzuki
et al., 2021). The latter is known to possess a nitrogen fixing cyanobacterial endosymbiont (UCYN-A;
Suzuki et al., 2021) that would have allowed it to adapt to the highly nutrient depleted environment. In
addition, *B. bigelowii* is a haptophyte that was found to perform extracellular calcification (Hagino et al.,
2016) and thus, may have benefited from the increase in the calcite saturation state.

Taxon-specific, optimum curve-like responses of phytoplankton growth to the combined effect of $H^+$ and $CO_2$ have previously been reported (Paul and Bach, 2020). $CO_2$ is usually considered the main source of carbon for primary production. However, most marine phytoplankton are capable of actively taking up bicarbonate thanks to carbon concentrating mechanisms (CCMs; Giordano et al., 2005; Price et al., 2008). Bicarbonate ions are accumulated in the cytosol, and later converted to $CO_2$ prior to carboxylation (Price et

al., 2008). In the present study DIC was increased according to an equilibrated (no reduction in $CO_2$) TA gradient. This manipulation was the main difference between the mesocosms. Thus, the bicarbonate availability levels attained in the intermediate treatments, where autotrophy was observed, may have been behind the detected peaks in production, alongside the potential relief of nutrient limitation explained above. Meaning that, in the current experiment, a certain nanoplankton species with more evolved CCMs

may have benefited from the higher DIC concentrations, and slight pH increase, directly and indirectly respectively.

Chi et al. (2014) studied different strains of microalgae and cyanobacteria as candidates for bicarbonate-based carbon capture for algae production (BICCAPS). Depending on the species, different growth rates and thresholds, and in some cases growth inhibition, were observed when these were cultured under varying

bicarbonate concentrations. This is likely due to species-specific ionic strength tolerance, meaning their capacity to adapt and thrive in varied bicarbonate ion concentrations, potentially explaining the observed threshold.

However, whether the peaks observed in $\Delta1500$ and $\Delta1800$ $\mu mol \cdot L^{-1}$, that drive the detected optimum curve (Table 2 and Figure 7) occurred by random chance and were thus not caused by the carbonate

chemistry conditions, remains unclear. In fact, when removing $\Delta1500$ and $\Delta1800$ $\mu mol \cdot L^{-1}$ from the model instead of the two highest treatments, these positive significant relationships with DIC vanish (Table 2). Although, these nutrient-decoupled peaks in production only occurred in mesocosms where TA, DIC and, to a lesser extent pH, were increased. It is a novel sighting since a response of this magnitude has not been observed in previous experiments carried out under nutrient depleted conditions, and/or while testing ocean

acidification, in the Canary Islands (Paul, et al., 2024: in prep). Consequently, and also considering the results from the cross-validation test, additional studies simulating the gradient applied here or similar, though with replicates, could further elucidate if such a threshold and the positive relation found below it hold. If the latter were further supported, long-term consequences, in terms of microbial community metabolic functioning associated with said changes, would need to be taken into consideration and further

evaluated before OAE implementation.

### 4.2 Challenges and limitations of OAE studies

A limitation of this experimental set up that should also be mentioned is that mesocosm studies are limited to temporal scales of weeks to months, precluding the study of potential longer-term effects. Additionally, secondary precipitation in the highest treatment likely occurred due to the substrate for nuclei formation provided by the mesocosm walls themselves, although it may not be the sole cause.

Hartmann et al. (2023) carried out $CO_2$ equilibrated alkalinity additions of up to $\Delta 2400$ µmol L$^{-1}$ using the same stock solutions as in our experiment. Biotic incubations that included phytoplankton and particles smaller than 55 µm, which are potential seed surface for nucleation, were set up. They observed no precipitates forming on the bottle walls, and thus no TA consumption, after 4 days. However, in their long-term, up to 90-day alkalinity stability experiment, precipitation was observed in the "untreated mode" (or control, meaning no particle addition), 10 days after the TA increase. Hartmann et al. (2023) hypothesized that precipitation was potentially triggered by the wall-effect since it was an abiotic treatment containing no particles larger than 0.2 µm. Furthermore, when precipitates from other experiments were added to the "treated mode" treatments, immediate and persistent precipitation was observed for both $\Delta 2100$ and $\Delta 2400$ µmol L$^{-1}$. Additionally, Wurgaft et al., (2021) found that TA loss via abiotic precipitation occurred at lower levels in a natural system than in these experiments due to the sediment particles in river plumes.

Thus, the secondary precipitation observed in the present study, as previously stated, may have been due to a combination of the wall-effect, including cleaning procedures that caused resuspension of particles present on the walls, but also to the existence of particles and cells in the water column. Whether carbonate formation would occur around the levels (~4500 µmol · L$^{-1}$) observed in this study in a natural oligotrophic, open ocean environment is still unclear. Actually, the theoretical aragonite saturation ($\Omega_{Ar}$) threshold of 12.5 above which carbonate precipitation was expected to occur (Morse and He, 1993), was never surpassed (Table 1).

### 4.3 Implications for future OAE research

Further experimental research at this scale is essential to test the effects of non-equilibrated OAE approaches as well. These may be more viable considering the current infrastructure since large scale equilibrated OAE application may require the use of reactors to $CO_2$ equilibrate the alkaline solutions prior to addition (Hartmann et al., 2023).

At the alkalinity point source, and depending on the alkalinity dispersal plume dynamics, the carbonate system perturbations associated to non-equilibrated OAE can be much stronger. Alkalinity loss would also

be triggered at much lower levels than those observed for $CO_2$ equilibrated OAE (Hartmann et al., 2023). Besides, when precipitation is triggered, a process by which precipitation keeps progressing past reaching the aragonite saturation levels of 12.5-13.5, and even ambient levels, also known as "runaway precipitation" (Moras et al., 2022), may be induced.

The findings of the current study suggest that carbonate-based, $CO_2$ equilibrated OAE may be environmentally safe in terms of the metabolic processes measured here, in an oligotrophic environment, even if abiotic precipitation were triggered. Although further research is required on the impacts of this phenomenon on other processes, i.e. on particle sinking due to ballasting. Moreover, uncertainty remains in the determination of responses to longer term exposure to the conditions simulated in this study, and in the levels at which abiotic precipitation may occur in the natural open ocean.

Several risks and co-benefits have been listed for this NET (Bach et al., 2019), although none have been really tested at a reasonable scale. This study concludes there may be a potential co-benefit to the addition of carbonates in solution, with $CO_2$ equilibration, where biological carbon sequestration is increased up to a certain threshold. Moreover, and as is true for ocean acidification, this response is species/group specific. In addition, past the mentioned threshold, production decreased but rates were comparable to those measured for the control and $\Delta 300$ µmol $\cdot$ L$^{-1}$ treatments. Therefore, no impact of equilibrated OAE past the ~4000 µmol $\cdot$ L$^{-1}$ TA threshold, and of abiotic precipitation at ~4300 µmol $\cdot$ L$^{-1}$, on the measured metabolic rates can be inferred.

## 5. Conclusions

An ideal Ocean Alkalinity Enhancement (OAE) deployment scenario was simulated under natural conditions. Total alkalinity (TA) was increased without the introduction of potentially harmful dissolution by-products, and $CO_2$ was chemically sequestered prior to the TA manipulation. The OAE approach employed within the specified TA range did not pose a threat to the pelagic microbial community in relation to the parameters monitored in the current study. Importantly, this held true even when abiotic precipitation occurred in the highest treatment. In fact, a potential co-benefit in the form of increased microbial community and primary production up to specific threshold. This increase could be driven either indirectly by the rise in pH, enhancing nitrogen cycling and consequently inorganic nutrient availability, or by the carbonate chemistry conditions, specifically increased Dissolved Inorganic Carbon (DIC) availability. Our discovery of a non-linear, optimum curve-like response in microbial production rates to the applied Dissolved Inorganic Carbon (DIC) gradient (as shown in Table 2) is noteworthy. This finding is novel and

warrants further investigation. Therefore, considering the substantial climatic benefits it could offer, additional research on carbon uptake efficiency and the effects of $CO_2$, but also non-$CO_2$ equilibrated OAE on natural microbial communities is of high priority.

**Data availability**

Datasets of the response variables presented in this study can be found in an online repository. The name of the repository is PANGAEA, and the access link is https://doi.pangaea.de/10.1594/PANGAEA.964537 (Marín-Samper et al., 2024). Biogeochemical data (nutrient concentrations and carbonate chemistry) will be made available in the same repository, without undue reservation.

**Author contributions**

Experimental concept and design: UR and JA. Execution of the experiment: All authors. Data analysis: LMS with input from NHH and JO. Original draft preparation: LMS. Review and editing: All authors.

**Financial support**

This research has been supported by the Horizon 2020 Research and Innovation Programme project OceanNETs ("Ocean-based Negative Emissions Technologies – analysing the feasibility, risks and co-benefits of ocean-based negative emission technologies for stabilizing the climate", grant no. 869357), and by the Helmholtz European Partnering project Ocean-CDR ("Ocean-based carbon dioxide removal strategies", Project No.: PIE-0021). Additional funding was provided through the EU H2020-INFRAIA's project AQUACOSM ("AQUACOSM: Network of Leading European AQUAtic MesoCOSM Facilities Connecting Mountains to Oceans from the Arctic to the Mediterranean", Project No.: 731065). This work was co-financed by the "Agencia Canaria de Investigación, Innovación y Sociedad de la Información" (ACIISI) of the "Consejería de Economía, Conocimiento y Empleo", and by the "Fondo Social Europeo (FSE) Programa Operativo Integrado de Canarias 2014-2020, Eje 3 Tema Prioritario 74 (85%)".

**Competing interests**

The authors declare that they have no conflict of interest.

**Acknowledgements**

The authors are grateful for the entire KOSMOS team of GEOMAR for all the logistical and technical work associated to the mesocosm campaign, coordinating all the on-site research activities, and for promoting fair data management and exchange. A special thank you goes to the biological oceanography group (GOB-ULPGC), in particular to Acorayda González, for helping with the oxygen measurements, and to Minerva Espino, Aja Trebec, Beatriz Fernández, Lucía Palacios, and Maria Fernanda Montero for carrying out a large volume of sample analyses. Also, we would like to acknowledge Levka Hansen (GEOMAR) for helping with the primary production through 14C uptake measurements, Julieta Schneider (GEOMAR) for the carbonate chemistry measurements, and Allanah Paul for the interesting discussions on data interpretation. Finally, we want to thank the Oceanic Platform of the Canary Islands (PLOCAN) and the University of Las Palmas of Gran Canaria (ULPGC) for providing all the essential facilities to conduct this experiment.

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
