# Peer review of "Assessing the impact of CO2 equilibrated ocean alkalinity enhancement on microbial metabolic rates in an oligotrophic system"

_EGUsphere, 2023_

## Editor Comment (EC1)

**General comments:**

The authors address the effects of increasing $CO_2$ equilibrated ocean alkalinity on marine microbes in an oligotrophic system. This study is timely since there has been increasing insight that negative emissions will be required to avoid dramatic climate change. Therefore, studying the potential effects of these carbon removal strategies is essential to informed decisions about their efficacy and practicality.

The manuscript is well design and, overall, well written and provides an important step to the knowledge of the effects of ocean alkalinity enhancement. Therefore, I recommend this paper for publication in *Biogeosciences*, after minor revisions.

**Specific comments:**

**Title**
The title of the manuscript puts emphasis on testing an oligotrophic system, but not much relevance was given in the introduction and discussion sections to the importance and particularities of these oligotrophic regions. Moreover, information concerning the nutrient concentrations during the experiment is not evident, since the Supplementary Figures are not cited in the manuscript (please correct this).

**P. 1, lines 5-** Correct formatting of the coma after Stephen D. Archer.

**Introduction**
**P. 2, line 56:** The citation "National Academies of Sciences, Engineering and Medicine, (2021) is not in the reference list.

**P. 2, line 72**: The citation "Renforth & Henderson (2017)" is not in the reference list.

**P.3, line 97:** What is meant by "first step"? A few other studies have considered the effects of ocean alkalinity enhancement. Please improve phrasing for clarity.

**Material and Methods**
**P. 4, line 122:** Please add n and Standard Deviation or Standard Error associated with the presented averages.

**P. 4, lines 123 to 125**: To calculate the carbonate system it is also necessary to know phosphate and silicate concentrations, please add the values used and refer to other potential publication or refer to Figure S5 in Supplementary Material.

**P. 4, line 125**: The citation "Uppström (1974)" is not in the reference list.

**P. 4, line 126**: The salinity used to calculate the carbonate system before and after the manipulation of total alkalinity was the same? Please add the difference caused by the manipulation and justify for not using the specific salinities.

**P. 4, line 129**: The word "(italics)" is unnecessary.

**P. 4, line 131**: Specify the pH scale and format "p" of "$p$CO2" to italic (throughout the manuscript).

**P. 4, line 134**: The reference to "pseudo random order" needs further clarification, since this term is used when an algorithm is applied to produce sequences of random numbers.

**P. 4, lines 141 to 143**: Considering the novelty of the study, it is important to provide more detailed information on the accuracy of the alkalinity manipulation, such as, effects related to increasing TA levels, the time between sampling and measurements, and potential precipitation effects.

**P. 5, line 152**: The citation "Bryan et al. (1976)" is not in the reference list.

**P. 5, line 168**: The collection of an integrated 2.5 m sample provides information of the communities that occur from 0 to 2.5 m depth. If the referred light intensity range was measured below the screen, the communities were exposed to very high light intensities. Hence, it would be useful to state the time frame to which the organisms were exposed to this high light intensity ($\sim$2300 µmol m$^{-2}$ s$^{-1}$). Furthermore, elaborate on the choice of the screen.

**P. 6, line 187**: The citation "Cermeño et al. (2012)" is not in the reference list.

**P. 6, lines 192 to 193**: This sentence should appear before, perhaps in line 189.

**P. 7, line 238**: It would be useful to have information about the groups that were defined with flow cytometry, in the Material and Methods section. This will facilitate the discussion.

**P. 7, line 250**: The citation "Wickham (2016)" is not in the reference list.

**Results**
**P. 7, line 255**: The graphs might be misleading in relation to the end of the so-called stable period, is it day 20 or 21? Moreover, with the exception of the highest treatment, where precipitation occurred, both TA and DIC were stable until the end of the experiment according to Figure 1. Please elaborate.

**P. 8, line 272**: Please adjust the graphs to the same dimensions to facilitate comparison and add a break symbol to call attention that the axis is not starting with 0. The legend of the X axis of this Figure and others would improve by modifying it to Time (d) or Elapsed time (d).

**P. 8, line 277**: Improve phrasing for clarity.

**P. 9, line 284**: Improve the figure caption of Figure 2 by adding "the temporal development" after "… oxygen production and consumption showing…" and removing it from the descriptions of the graphs A, B, C and D.

**P. 10, line 289**: Chlorophyll *a* should be represented with the "a" in italic here and throughout the manuscript.

**P. 10, line 291**: The authors refer to "differences", are these differences statistically significant? Please add information.

**P. 11, line 295**: Improve phrasing for clarity.

**P. 11, lines 298 to 300**: If the data was not collected, what does the data point refer to? Improve phrasing for clarity.

**P. 11, line 300**: Improve figure caption for clarity.

**P. 11, line 305**: Could be useful to present a graph to support the statement.

**P. 12, line 312/3**: Considering that the manuscript is testing potential relationships in relation to TA enhancement it would be useful to either have an additional axis showing TA or indication in the text of the reasoning behind using DIC (more biologically relevant).

**P. 12, line 312/3**: Add space after Figure 5. Cell abundances of the nanoeukaryotes seem high for the region and nutrient / Chl*a* concentrations found, please re-check.

**P. 12, lines 315 to 316**: Add Figure number to Nanoeukaryote (1) and Nanoeukaryote (2). The text should be improved for fluidity, namely reference to the figure and graphs should be cited in the text and sequentially (change the order of the graphs in the Figure or in the text). Moreover, there should be an introductory sentence relating to the observed trends in primary production and metabolic balance. Finally, in the discussion section, the dominant species of certain data points are presented, but there is no information about the initial community.

**P. 13, lines 342 to 343**: Elaborate on the need to have the terms nanophytoplankton and nanoeukaryote.

**P. 14, lines 353 to 355**: Improve sentence.

**P. 16, lines 376 to 377**: Specify what portion of the community is considered in this sentence.

**P. 16, line 380**: Please add that 4500 µmol kg$^{-1}$ is the final TA concentration.

**P. 16, lines 382 to 384**: Improve phrasing and remember that there were significant differences in part of the community. Moreover, it is important to question, whether the small changes observed might have a long-term effect on the functioning of the microbial communities.

**P. 16, lines 386 to 388**: Care should be given when comparing oligotrophic versus eutrophic environments loosely, since communities vary seasonally, with consequences to the initial community.

**P. 17, line 406**: Which results? Do the authors have information to go into more detail than the group nanophytoplankton?

**P. 17, line 408**: The two studies considered tested different TA ranges. Therefore, one should compare within the range to which both have data for.

**P. 18, lines 421 to 423**: Please improve the sentence for clarity. What is meant with accumulation of inorganic nutrients? How is it related to the nitrogen cycle?

**P. 19, lines 461**: Elaborate on "…ion strength tolerance…".

**P. 20, line 496**: Citation (Morse and He, 1993) is not in the reference list. Please change "and" to "&" to uniformize formatting.

**P. 21, line 516**: Remove "; "from the citation (Bach et al., 2019).

**P. 21, Conclusions**: Despite the relevance of primary production differences, these are not referred in the conclusions, while "…minor changes in species composition…" are emphasized but the work focused on groups. Please improve the section accordingly.

*P refers to page. Line counts continue throughout the manuscript.*

---

## Author Comment (AC1)

Specific comments addressed:

**Title**

The title of the manuscript puts emphasis on testing an oligotrophic system, but not much relevance was given in the introduction and discussion sections to the importance and particularities of these oligotrophic regions. Moreover, information concerning the nutrient concentrations during the experiment is not evident, since the Supplementary Figures are not cited in the manuscript (please correct this).

Emphasis is given to carrying out the experiment in an oligotrophic system in the title because the goal was to simulate ocean liming. The latter has been generally discussed to consist of injecting alkalinity to the open ocean surface. Therefore, the Canary Islands were chosen due to their oligotrophic nature and resemblance to an open ocean system. This has been further clarified in the introduction (P2, line 74 and P3, line 103) and it was stated at the beginning of the discussion (P17, line 390). The nutrient concentration temporal development (Supp. Fig S1, before it was S2), has been cited in the introduction, in the methods section and in the discussion.

P. 1, lines 5- Correct formatting of the coma after Stephen D. Archer.

Corrected.

**Introduction**

P. 2, line 56: The citation "National Academies of Sciences, Engineering and Medicine, (2021) is not in the reference list.

Corrected.

P. 2, line 72: The citation "Renforth & Henderson (2017)" is not in the reference list.

Corrected.

P.3, line 97: What is meant by "first step"? A few other studies have considered the effects of ocean alkalinity enhancement. Please improve phrasing for clarity.

This comment has been addressed by specifying the scale of this experiment. Because this experiment is the first one at a mesocosm scale on ocean alkalinity enhancement, we the authors considered it wise to simulate a best-case scenario. Which is why this ("first step", changed to "first attempt") was specifically mentioned.

**Material and Methods**

P. 4, line 122: Please add n and Standard Deviation or Standard Error associated with the presented averages.

Corrected, standard errors were added to the table. We would like to thank the reviewer because another mistake was noticed. The averages in the previous version of the manuscript included the entire experiment. In this second version, the days prior to the addition were excluded from the calculations.

P. 4, lines 123 to 125: To calculate the carbonate system it is also necessary to know phosphate and silicate concentrations, please add the values used and refer to other potential publication or refer to Figure S5 in Supplementary Material.

We included a citation to the Supplementary Material Figure that portrays the temporal development of the nutrient concentrations. Due to this, the order of the supplementary figures was corrected. Now this supplementary figure is number S1.

P. 4, line 125: The citation "Uppström (1974)" is not in the reference list.

Corrected.

P. 4, line 126: The salinity used to calculate the carbonate system before and after the manipulation of total alkalinity was the same? Please add the difference caused by the manipulation and justify for not using the specific salinities.

Salinity throughout the study mildly increased due to evaporation although it remained overall quite constant. To convey the evaporation effect, after day 17 (which marks approximately the middle of the experiment) the salinity in said calculations was increased. It is true that the salinity increased slightly due to the addition of NaHCO3, and Na2CO3. In the corrected table, the portrayed averages were calculated based on non-normalized carbonate chemistry parameters. Meaning, the latter were calculated with the *in-situ* salinity values.

P. 4, line 129: The word "(italics)" is unnecessary.

Corrected.

P. 4, line 131: Specify the pH scale and format "p" of "pCO2" to italic (throughout the manuscript).

Corrected.

P. 4, line 134: The reference to "pseudo random order" needs further clarification, since this term is used when an algorithm is applied to produce sequences of random numbers.

We addressed this comment by changing "pseudo random" to "random" order. We also explained that the treatments were the ones arranged in random order, the mesocosms along the pier did go from 1 to 9.

P. 4, lines 141 to 143: Considering the novelty of the study, it is important to provide more detailed information on the accuracy of the alkalinity manipulation, such as, effects related to increasing TA levels, the time between sampling and measurements, and potential precipitation effects.

An overview publication (Paul et al., 2024) about this study that has been accepted for public discussionin the same special issue, is addressing this in more detail.

P. 5, line 152: The citation "Bryan et al. (1976)" is not in the reference list.

Corrected.

P. 5, line 168: The collection of an integrated 2.5 m sample provides information of the communities that occur from 0 to 2.5 m depth. If the referred light intensity range was measured below the screen, the communities were exposed to very high light intensities. **Hence, it would be useful to state the time frame to which the organisms were exposed to this high light intensity (~2300 µmol m-2 s-1). Furthermore, elaborate on the choice of the screen.**

The mesocosms used enclosed only the top 2.5 m of the water column. We chose this screen because it mimics the natural attenuation at roughly 1m depth and thus better represents the light conditions at approximately the middle of water column inside the mesocosms, compared to full direct daylight (i.e. surface values more or less). Natural light variability during the experimental period was high, which is why we chose to state the range of light intensity instead of the average value. We thought the range of exposure was more important to mention because it gives an idea of the natural light variability to which mesocosms, and incubators were exposed. The highest intensities (~2300 µmol m-2 s-1) were mostly observed between noon and 15:00 during the last week of the experiment (T25 onwards), and in other sporadic occasions, which is why we did not consider the need to state this.

P. 6, line 187: The citation "Cermeño et al. (2012)" is not in the reference list.

Corrected.

P. 6, lines 192 to 193: This sentence should appear before, perhaps in line 189.

Corrected.

P. 7, line 238: It would be useful to have information about the groups that were defined with flow cytometry, in the Material and Methods section. This will facilitate the discussion.

We thank the reviewer for their suggestion, the requested information was added to the methods section.

P. 7, line 250: The citation "Wickham (2016)" is not in the reference list

Corrected.

**Results**

P. 7, line 255: The graphs might be misleading in relation to the end of the so-called stable period, is it day 20 or 21? Moreover, with the exception of the highest treatment, where precipitation occurred, both TA and DIC were stable until the end of the experiment according to Figure 1. Please elaborate.

Vertical dotted lines mark the time of alkalinity addition on day 4 and the point at which phase-I transitions to phase-II on day 20. Yes, carbonate chemistry was stable in all treatments throughout except the highest one. The distinction between phase I and phase II was based on a change in the biological response, evident in many of the variables measured, rather than on a change in the alkalinity in any of the treatments. We agree, our original text did imply that the phase transition was based on the change in

alkalinity dynamics in the highest treatment – this is not the case and we have clarified this.

P. 8, line 272: Please adjust the graphs to the same dimensions to facilitate comparison and add a break symbol to call attention that the axis is not starting with 0. The legend of the X axis of this Figure and others would improve by modifying it to Time (d) or Elapsed time (d).

Corrected. This x axis nomenclature will be used in complementary publications thus we find it appropriate to leave it as is. However, it has been further clarified in all the temporal development figure captions by adding the following sentence at the end: The X-axis represents the number of days elapsed since the beginning of the experiment.

P. 8, line 277: Improve phrasing for clarity.

Corrected. The start of this section was changed to: Another reason for the delineation of the mentioned phases (I: days 5-19, II: days 21-33) is the observed increase in production and chlorophyll-a concentration in specific intermediate treatments after day 20 when compared to phase I (Figure 2). This division of the experimental period was chosen to facilitate the interpretation of the system's response.

P. 9, line 284: Improve the figure caption of Figure 2 by adding "the temporal development" after "… oxygen production and consumption showing…" and removing it from the descriptions of the graphs A, B, C and D.

Corrected.

P. 10, line 289: Chlorophyll a should be represented with the "a" in italic here and throughout the manuscript.

Corrected.

P. 10, line 291: The authors refer to "differences", are these differences statistically significant? Please add information.

No, they were not. This comment has been addressed by specifying that visually these differences occur, although they are not significant.

P. 11, line 295: Improve phrasing for clarity.

This comment was addressed by separating the sentence that starts on line 295 (now line 305) into two sentences.

P. 11, lines 298 to 300: If the data was not collected, what does the data point refer to? Improve phrasing for clarity.

The sentence was rephrased as:

(Now line 309) Data for PP on day 27, when oxygen production and Chl$a$ concentration in $\Delta$1800 reached the highest levels recorded throughout the entire experiment for all mesocosms, were not collected. Consequently, the peak in $\Delta$1800 reflected by Chl$a$ (Figure 4A and B), as well as the GCP and NCP rates (Figure 2), which surpassed those in $\Delta$1500, was excluded (Figure 4C and D).

P. 11, line 300: Improve figure caption for clarity.

Corrected. The figures were cited in sequential order within the caption. It was clarified that 'Chla' stands for total chlorophyll-a, and explicit specifications were added for the nano size range as well as the acronym 'PP,' which refers to primary production.

P. 11, line 305: Could be useful to present a graph to support the statement.

Since this article is not meant to focus on community structure changes much, we felt that the statistics alone were enough evidence. A complementary publication that is currently in preparation will likely provide this information.

P. 12, line 312/3: Considering that the manuscript is testing potential relationships in relation to TA enhancement it would be useful to either have an additional axis showing TA or indication in the text of the reasoning behind using DIC (more biologically relevant).

The reason behind using the DIC gradient in the analyses performed to test a non-linear response of the metabolic rates and Nano contributions and abundances, is stated in line 344 of the new manuscript ("TA and DIC, in an equilibrated OAE approach, vary together (as TA increases, so does DIC; Supp. Fig. S5) and, if a potential non-linear response between the metabolic parameters listed in Table 2 were to be considered, the driver behind these relationships would most likely be DIC (key substrate for carbon fixation; Badger et al., 1998), not TA").

P. 12, line 312/3: Add space after Figure 5. Cell abundances of the nanoeukaryotes seem high for the region and nutrient / Chla concentrations found, please re-check.

Thank you for noticing. Yes, they were too high. The calculations were revised, and we noticed that there was a mistake in the applied flow rate for the estimates. It has now been corrected.

P. 12, lines 315 to 316: Add Figure number to Nanoeukaryote (1) and Nanoeukaryote (2). The text should be improved for fluidity, namely reference to the figure and graphs should be cited in the text and sequentially (change the order of the graphs in the Figure or in the text). Moreover, there should be an introductory sentence relating to the observed trends in primary production and metabolic balance. Finally, in the discussion section, the dominant species of certain data points are presented, but there is no information about the initial community.

Figure numbers/letters were specified for the two nano populations separately in the caption. Furthermore, the sentence "The two latter correspond to two different nanoeukaryote populations" was added right after listing what each figure from A to D represents.

The order in which the figures in Figure 6 are cited in the text was changed (Figure 6A to 6D are sequentially referenced).

Two introductory sentences summarizing the results portrayed in the previous sections were added: "The second phase of the experiment was characterized by an increase in production and Chl*a* concentrations in all intermediate treatments below the two hishest and above the two lowest treatments, except $\Delta 1200$. While phase I was distinguished by

extremely low GP, NCP, PP rates and Chl*a* throughout and across all mesocosms. (line 329-332)"

Exemplary samples for microscopy had been analyzed at the time for those days to determine what species was growing. Data on the initial community, besides what is provided here, were not made available to us. These data will contribute to a complementary research publication addressing community structure changes specifically.

P. 13, lines 342 to 343: Elaborate on the need to have the terms nanophytoplankton and nanoeukaryote.

To prevent confusion with terminology, we adopted the terms picoplankton, nanoplankton, and microplankton when referring to the size fractions of Primary Production (PP) and Chlorophyll-a (Chla), avoiding the use of the term 'nanophytoplankton.' Specifically, the terms nanoeukaryotes (1) and (2) denote clusters of phytoplankton identified within the nanoplankton fraction. These clusters encompass both autotrophic and mixotrophic flagellates, with the latter corresponding to larger cells exhibiting yellow fluorescence. Another reason for the use of different terminology to refer to these two datasets was to distinguish them when referenced.

P. 14, lines 353 to 355: Improve sentence.

Corrected. The end part of the sentence was placed at the beginning.

**Discussion**

P. 16, lines 376 to 377: Specify what portion of the community is considered in this sentence.

Corrected.

P. 16, line 380: Please add that 4500 µmol kg-1 is the final TA concentration.

Corrected.

P. 16, lines 382 to 384: Improve phrasing and remember that there were significant differences in part of the community. Moreover, it is important to question, whether the small changes observed might have a long-term effect on the functioning of the microbial communities.

The reviewer raises a good point, the long-term effect evaluation represents a limitation in many studies, including mesocosm experiments like the one conducted here. This aspect is explicitly addressed in the opening sentence of the "4.2 Challenges and Limitations…" discussion section. However, since the slight response observed is novel and disappears by the end of the study, this aspect is not further discussed because it may be based on too many assumptions. Due to this and other reasons, we conclude that further research is required. Nonetheless, we have added a sentence at the end of section 4.1 addressing it.

P. 16, lines 386 to 388: Care should be given when comparing oligotrophic versus eutrophic environments loosely, since communities vary seasonally, with consequences to the initial community.

The text included in this portion of the discussion alludes to pending hypotheses since OAE has been hardly studied to this point. Based on said hypotheses, we argue that nutrient limitation may have concealed a clearer/stronger response, and we wanted to emphasize key knowledge gaps that remain. The goal thus was not necessarily to compare eutrophic versus oligotrophic, since we agree these terms do not only describe the environmental conditions of a system, but also the community it can sustain, which are both highly variable with time. It was to point out some hypotheses about biotic responses in eutrophic systems in support of the idea that, in the current experiment, the highly oligotrophic conditions could have limited the community's response. We have added a sentence in this section explicitly stating that research on OAE impacts in oligotrophic systems at a comparable scale does not exist. Right before mentioning eutrophic environments.

P. 17, line 406: Which results? Do the authors have information to go into more detail than the group nanophytoplankton?

We have now clarified which results are being referred to.

P. 17, line 408: The two studies considered tested different TA ranges. Therefore, one should compare within the range to which both have data for.

We thank the reviewer for this insightful comment. We addressed it by comparing the results from our study with Ferderer, et al.'s (2022) in more detail. We found a mistake in the result interpretation carried out for the latter study. Therefore, we discuss this point differently. We hope the reviewer considers the new text to be appropriate and well discussed.

P. 18, lines 421 to 423: Please improve the sentence for clarity. What is meant with accumulation of inorganic nutrients? How is it related to the nitrogen cycle?

The first portion of the comment was addressed by changing "accumulation of inorganic nutrients" to "an increase of inorganic nutrients in relation to the TA manipulation, potentially caused by enhanced nutrient cycling". We believe that a possible explanation for the observed long-term response is that the slight increase in pH favored heterotrophic organic nitrogen turnover. We state this a few lines down (P19, line 433) after relating our results with those from Paul, et al (2024), which is a publication about the same study that is in preparation. We find that comparing their results with ours is key to providing a potential explanation for the observed response in production and its timing. We also added a citation to Sup. Fig S2 where nutrient concentrations are provided and NOx clearly increases in the second phase which would further support the aforementioned explanation.

P. 19, lines 461: Elaborate on "…ion strength tolerance…".

To address this comment, we extended this sentence changing it to: "This is likely due to species-specific ionic strength tolerance, indicating their capacity to adapt and thrive in varied bicarbonate ion concentrations, potentially explaining the observed threshold"

P. 20, line 496: Citation (Morse and He, 1993) is not in the reference list. Please change "and" to "&" to uniformize formatting.

Corrected.

P. 21, line 516: Remove "; "from the citation (Bach et al., 2019).

Corrected.

P. 21, Conclusions: Despite the relevance of primary production differences, these are not referred in the conclusions, while "…minor changes in species composition…" are emphasized but the work focused on groups. Please improve the section accordingly.

We thank the reviewer for this comment and fully agree. We removed the part about minor changes in species composition since, as mentioned above, it is not the main focus of this study. Then changed the following portion of the conclusions to: "…In fact, we observed a potential co-benefit in the form of increased microbial community and primary production up to a specific threshold. This increase could be driven either indirectly by the rise in pH, enhancing nitrogen cycling and consequently inorganic nutrient availability, or by the carbonate chemistry conditions, specifically Dissolved Inorganic Carbon (DIC) availability., our discovery of a non-linear, optimal curve-like response in microbial production rates to the applied DIC gradient (as shown in Table 2) is noteworthy. …"

P refers to page. Line counts continue throughout the manuscript.

**References**

Badger, M. R., Andrews, T. J., Whitney, S. M., Ludwig, M., Yellowlees, D. C., Leggat, W., & Price, G. D. (1998). The diversity and coevolution of Rubisco, plastids, pyrenoids, and chloroplast-based CO2-concentrating mechanisms in algae. *Canadian Journal of Botany*, *76*(6), 1052–1071. https://doi.org/10.1139/b98-074

Paul, A. J., Haunost, M., Goldenberg, S., Sanchez, N., Schneider, J., Suitner, N., & Riebesell, U. (2024). *Ocean alkalinity enhancement in an open ocean ecosystem: Biogeochemical responses and carbon storage durability. March*, 1–31.

---

## Author Comment (AC2)

We would like to extend our heartfelt gratitude for the reviewer's thoughtful feedback regarding our manuscript. The time and commitment conferred to providing constructive criticism are truly appreciated. We believe the suggestions made have played a decisive role in refining the content of our manuscript, and we have incorporated all the specific revisions to improve its clarity and coherence. Should there be any elements that warrant additional clarification, we welcome the opportunity for continued discussion.

Lines 82-85: "Additionally, the type…" –This sentence is a bit convoluted; it also needs additional citations. For example, I don't think including "biogenic" when citing Moras et al. 2022 is accurate as the referenced study examines runaway $CaCO_3$ precipitation when using proposed ocean-liming minerals (e.g., CaO and Ca[OH]$_2$) – and they hypothesize that precipitation likely occurred on the surface of the undissolved ocean-liming minerals used in their study; while this offers support to abiotic particles affecting OAE efficacy – at least when using the noted ocean-liming minerals – it seems the authors are also implying that the presence of biogenic $CaCO_3$ could induce similar precipitation. If so, additional citations should be included. Citations for the other potential influences on OAE efficiency noted in this sentence would also be helpful.

We thank the reviewer for this insightful comment, and we agree. Two new references providing support for precipitation occurring in the presence of particles of biogenic origin have been added right after the word "biogenic". The sentence the reviewer considers convoluted has been divided in two.

Line 120: What the authors mean by "air-equilibrated…stock solutions" is not entirely clear here (e.g., did they bubble their solutions?). Although further explanation is provided later in the manuscript, a brief explanation along the lines presented in Federer et al. (2022) would provide clarity early on. Additionally, the method of alkalinity addition being simulated in this study is not entirely clear: if the goal was to simulate an ocean-liming scenario (as stated elsewhere in the paper – e.g., line 370), why weren't calcium concentrations also increased? Would these results also apply to aqueous hydroxide addition? For example, the use of NaOH – after its reaction with seawater – is effectively alkalinity enhancement through sodium carbonate addition. The authors also use "carbonate-based" (line 370) to describe the form of alkalinity addition being simulated – would these results apply to the use of other carbonate minerals (e.g., dolomite) as well?

Yes, agreed, this is confusing. Air-equilibrated solutions refers to them containing carbon in accordance with the targeted TA level. The order of this sentence was changed by moving air-equilibrated to before alkalinity gradient. Thus, attributing air-equilibration to the nature of the alkalinity addition, rather than the solutions themselves, which were not bubbled.

Sure, the calcium concentrations would need to be increased. However, and as a first step in the direction of evaluating impacts of ocean liming (meaning since no previous work had been carried out at the time), only alkalinity was increased. Adding calcium to the system, instead of or together with Na, would have been a confounding variable that, with no prior information would have complicated the system's response interpretation.

In this study equilibration was attained by adding carbon through the Na2CO3 and NaHCO3 solutions in proportion. This brings me to answering the two last questions. We do not think our results apply to an addition of aqueous hydroxide because the specific chemical pathways are different. It may be comparable if after the addition, thorough bubbling is undertaken to ensure pCO2 level restoration. We believe though this study provides a baseline that will help the interpretation of results from other experiments set out to increase alkalinity together with Ca or CaMg (like in the set-out example, dolomite), or basically simulating any other specific carbonate mineral additions.

Line 295: The spike in GP:CR in the Δ1200 treatment (Figure 2D) just before phase II is interesting – especially as differences were seen in the contribution of the micro size class and PER% during phase II for this treatment relative to other treatments (Figures 3 and 5). Do the authors have a hypothesis as to what may have caused the spike in this treatment?

We thank the reviewer for pointing out the spike in GP / NCP in the Δ1200 treatment. With the available datasets, we have not been able to explain why this treatment behaved differently. Microplankton community composition and structure is the focus of a complementary publication that is currently in preparation. They will hopefully be able to better address the difference between the D1200 and the rest of the treatments.

Line 314 and throughout: Using "community composition" seems a bit misleading as we can't deduce how relative species abundances or phytoplankton functional group (PFG) relative abundances might have been affected within each size class, especially seeing as how there is PFG overlap among size classes (e.g., Pierella et al. 2020). As such, stating that "only minor changes in species composition were observed" (line 531) seems a bit premature. If the authors wish to use "community composition" to describe their results, they should note that potential changes in the relative abundances of species or PFGs within size classes might be masked.

We want to thank the reviewer and we agree. The title of the section was changed to Pico- and Nano- eukaryote abundances.

Lines 315-316: It's not clear what criteria were used to differentiate nanoeukaryote (1) and nanoeukaryote (2) populations. Figures 3 and 4 have the nano community as one group, but two populations are discussed in Section 3.3 and presented in Figure 6.

Further details describing this differentiation have been added to section 3.3. Figures 3 and 4 refer to size-fractionated 14C and Chla results. Since both Nano populations could be included within the Nano size range, no differentiation can be inferred using the two latter datasets.

Minor:

Line 51: "Process that is…" – this sentence is not complete and would flow better if it were joined with the previous sentence.

Amended

Lines 54 and 59: Should "carbon dioxide removal" and "negative emissions technologies" be capitalized here?

Amended

Line 55 and throughout: "…hard to abate emissions…" should be written as "…hard-to-abate emissions…". Here – and throughout the manuscript (e.g., line 61: "…carbonate- or silicate-based alkaline…") – phrasal adjectives are often incorrectly written.

Amended

Line 175: Incubation time is represented as "$h_D$" and "$h_L$" in the two equations rather than "T" as noted in the text (line 182).

Amended

Line 187: I couldn't find Carmeño et al. 2012 in the reference list – check that all references are included.

Amended

Line 284 and following: Panel D in Figure 2 is labeled "E" in the caption. GP:CR is also shown as "GCP over CR". The authors should verify that figure captions match plot labels and in-text references.

Line 290: The axes in Figure 3 are difficult to read.

Amended. The size of the figure was increased to fit a whole page.

Line 400: At the beginning of this section, the authors' use of "these results" make it a bit difficult to determine to which study they are referring. For example, is the sentence beginning at the end of line 405 referring to this study or the Ferderer et al. 2022 study?

Amended.

Line 410: Why is Figure 7 presented here instead of in the Results section?

Amended. The figure has been cited and moved up to the "3.4 Non-linear response vs no response"

Line 420: The sentence beginning with "In addition…" should be combined with the previous sentence.

Amended.

References

Ferderer, A., Chase, Z., Kennedy, F., Schulz, K. G., & Bach, L. T. (2022). Assessing the influence of ocean alkalinity enhancement on a coastal phytoplankton community. *Biogeosciences, 19(23), 5375–5399.*

Pierella Karlusich, J. J., Ibarbalz, F. M., and Bowler, C. (2020). Phytoplankton in the Tara ocean. *Annu. Rev. Mar. Sci., 12, 233–265.*

---

## Author Response (AR2)

**Response to RC2 and editor-in-chief comments**

We want to thank both the Referee #2 and the editor, Lydia Kapsenberg, for their time and valuable feedback. The specific revisions suggested by Referee #2 have been amended. Additionally, ocean liming has been removed from the text altogether, and it has been substituted with carbonate-based OAE.